# Molecular insights into the endoperoxide formation by Fe(II)/α-KG-dependent oxygenase NvfI

Takahiro Mori [1,2,3,6 ✉], Rui Zhai[1,6], Richiro Ushimaru[1,2,4], Yudai Matsuda [5] & Ikuro Abe [1,2 ✉]

Endoperoxide-containing natural products are a group of compounds with structurally unique cyclized peroxide moieties. Although numerous endoperoxide-containing compounds have been isolated, the biosynthesis of the endoperoxides remains unclear. NvfI from *Aspergillus novofumigatus* IBT 16806 is an endoperoxidase that catalyzes the formation of fumigatonoid A in the biosynthesis of novofumigatonin. Here, we describe our structural and functional analyses of NvfI. The structural elucidation and mutagenesis studies indicate that NvfI does not utilize a tyrosyl radical in the reaction, in contrast to other characterized endoperoxidases. Further, the crystallographic analysis reveals significant conformational changes of two loops upon substrate binding, which suggests a dynamic movement of active site during the catalytic cycle. As a result, NvfI installs three oxygen atoms onto a substrate in a single enzyme turnover. Based on these results, we propose a mechanism for the NvfI-catalyzed, unique endoperoxide formation reaction to produce fumigatonoid A.

[1] Graduate School of Pharmaceutical Sciences, The University of Tokyo, Bunkyo-ku, Tokyo, Japan. [2] Collaborative Research Institute for Innovative Microbiology, The University of Tokyo, Bunkyo-ku, Tokyo, Japan. [3] PRESTO, Japan Science and Technology Agency, Kawaguchi, Saitama, Japan. [4] ACT-X, Japan Science and Technology Agency, Kawaguchi, Saitama, Japan. [5] Department of Chemistry, City University of Hong Kong, Kowloon, Hong Kong SAR, China. [6] These authors contributed equally: Takahiro Mori, Rui Zhai. ✉email: tmori@mol.f.u-tokyo.ac.jp; abei@mol.f.u-tokyo.ac.jp

Endoperoxides are structurally unique heterocycles with a characteristic peroxide moiety. They are present in a variety of natural products, such as terpenoids, alkaloids, polyketides, and meroterpenoids, produced in plants, fungi, and other organisms[1–3]. Due to the high reactivity of the endocyclic peroxide bond, these compounds display interesting biological properties, including antimalarial, antitrypanosomal, antibacterial, and antitumor activities[4,5].

Despite the promising biological properties of the endoperoxide-containing natural products, our knowledge regarding the enzymatic mechanism of endoperoxide formation remains limited. Presently, only two types of enzymes, the heme-dependent prostaglandin H synthase (also known as cyclooxygenase, COX) and the non-heme Fe(II)- and α-ketoglutarate (α-KG)-dependent dioxygenase fumitremorgin B endoperoxidase FtmOx1 from *Aspergillus fumigatus* have been mechanistically analyzed in detail[6–10] (Supplementary Fig. 1). In both cases, catalytic Tyr residues play key roles in the endoperoxide-forming reactions. In the reaction catalyzed by COX, a Tyr residue is oxidized by the ferryl species of the heme cofactor to generate a tyrosyl radical, which then abstracts a hydrogen atom from the arachidonic acid substrate. The resulting substrate radical reacts with molecular oxygen to generate a peroxyl radical intermediate. A subsequent cyclization reaction via radical addition across a double bond, and a reaction with another molecular oxygen produce the second peroxyl radical intermediate, which is eventually quenched by hydrogen atom transfer from the Tyr residue to afford the prostaglandin G2 product. For the FtmOx1 reaction, Yan et al. proposed a COX-like radical mechanism in which a tyrosyl radical, developed from the reaction of Tyr224 with a ferryl species, abstracts a hydrogen atom from C-21 of fumitremorgin B[9]. The resulting peroxyl radical transfers to the C-26-C-27 double bond, and the radical at C-26 is reduced by Tyr224 to generate verruculogen. In contrast, a recent detailed study on the reaction mechanism of FtmOx1 by Bollinger and co-workers proposed a distinct mechanism[10] in which the ferryl species directly abstracts the hydrogen atom from fumitremorgin B, and Tyr68, instead of Tyr224, serves as the H• donor to the C-26-centered radical intermediate, resulting in the product formation.

Fumigatonoid A (**1**), an endoperoxide-containing fungal meroterpenoid, is a biosynthetic intermediate of novofumigatonin, discovered in *Aspergillus novofumigatus* IBT 16806[11]. Our biosynthetic analysis of novofumigatonin revealed that the Fe(II)/α-KG-dependent enzyme NvfI accepts asnovolin A (**2**) as a substrate and introduces three oxygen atoms, including an endoperoxide bridge and C-3'-hydroxyl group, to generate fumigatonoid A (Fig. 1).[12] Sequence and phylogenetic analyses of the NvfI homologs and characterized Fe(II)/α-KG-dependent enzymes in secondary metabolite biosynthesis indicated that NvfI shares low sequence similarity (10–15% amino acid identity) and belongs to a different clade from FtmOx1 (Supplementary Fig. 2)[9,10,13–39]. These observations imply that the mechanism of the endoperoxide-forming reaction of NvfI is different from that of FtmOx1. In particular, the mechanism by which NvfI prioritizes the endoperoxide-forming reaction but not hydroxylation, the usual transformation catalyzed by α-KG-dependent non-heme iron enzymes, remains enigmatic.

Here, we report structural and functional analyses of NvfI. The structural analysis and mutagenesis studies revealed that the reaction of NvfI is unlikely to involve tyrosyl radical, in contrast to the COX and FtmOx1 mechanisms. Instead, we propose that the dynamic conformational changes of NvfI enable the repositioning of the substrate radical intermediate in the active site, thus evading the biosynthetically undesired hydroxylation and resulting in the reaction with molecular oxygen.

## Results

**In vitro characterizations of NvfI.** The in vitro enzyme reaction of NvfI with asnovolin A (**2**) generated fumigatonoid A (**1**) and its isomer **3**, which is produced via iron-mediated, non-enzymatic isomerization of **1**, as dominant products using α-KG, and $O_2$ as co-substrate (Fig. 2a)[12]. To confirm the stoichiometry of the co-substrates, in vitro enzyme reactions were performed using varied ratios of **2** and α-KG under aerobic conditions. As a result, **2** was almost fully consumed when the α-KG: **2** ratio reached *ca.* 1: 1 (Fig. 2b and Supplementary Fig. 3a), which suggested that one equivalent of α-KG was consumed during the formation of **1**. On the other hand, for measurement of oxygen stoichiometry, enzyme reactions were performed using different amount of $O_2$. As a result, the production of **1** increased with increasing $O_2$ concentrations, and reached plateau when the ratio was *ca.* 2 (Supplementary Fig. 3b, c), which indicated that two equivalents of $O_2$ were required for the installation of three oxygen atoms onto **2**.

Next, to analyze the origin of the oxygen atoms in **1** generated by NvfI, we performed labeling experiments using $^{18}O_2$ and $H_2^{18}O$ (Fig. 2c–g). When **2** was incubated with NvfI under $^{18}O_2$ (98 atom $^{18}O$%), a major +4 peak (*m/z* 499) of **1** was observed in the production of **1** (*m/z* 501 (+6): *m/z* 499 (+4): *m/z* 495 (0) = 1.2%: 95.7%: 2.0%) (Fig. 2 and Supplementary Table 1). Interestingly, in the experiment with $H_2^{18}O$ (final concentration 78 atom $^{18}O$%), a major +2 (*m/z* 497) peak was observed, along with a minor peak of the unlabeled **1** and +4 (*m/z* 499 (+4): *m/z* 497 (+2): *m/z* 495 (0) = 17.0%: 62.5%: 20.4%) (Fig. 2 and Supplementary Table 1). Furthermore, the double labeling with 78% $H_2^{18}O$ and 98% $^{18}O_2$ generated +4 (*m/z* 499), +6 (*m/z* 501) and +8 (*m/z* 503) products of **1** (*m/z* 503 (+8): *m/z* 501 (+6): *m/z* 499 (+4): *m/z* 497 (+2): *m/z* 495 (0) = 23.2%: 56.7%: 18.3%: 1.8%: 0.4%). Although three oxygen atoms should be newly installed in a reaction cycle, these results indicated that at most four oxygen atoms, two from $O_2$ and two from solvent water, were incorporated into **1**.

Considering the structure of **1**, the $O_2$ molecule is likely incorporated into the endoperoxide moiety in **1**. An oxygen atom from water should be incorporated as the carbonyl oxygen at C-4',

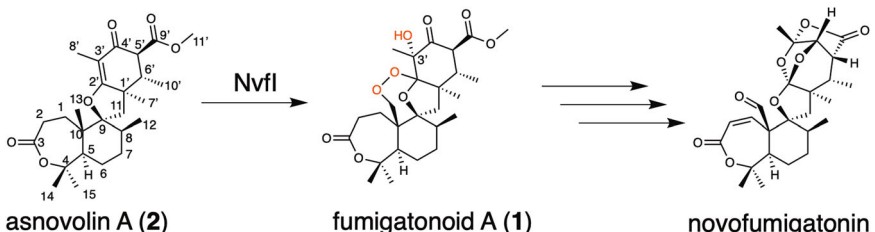

**Fig. 1 The enzyme reaction of NvfI.** NvfI catalyzes the production of fumigatonoid A (**1**) from asnovolin A (**2**). Subsequent modification reactions by other biosynthetic enzymes generate the final product novofumigatonin.

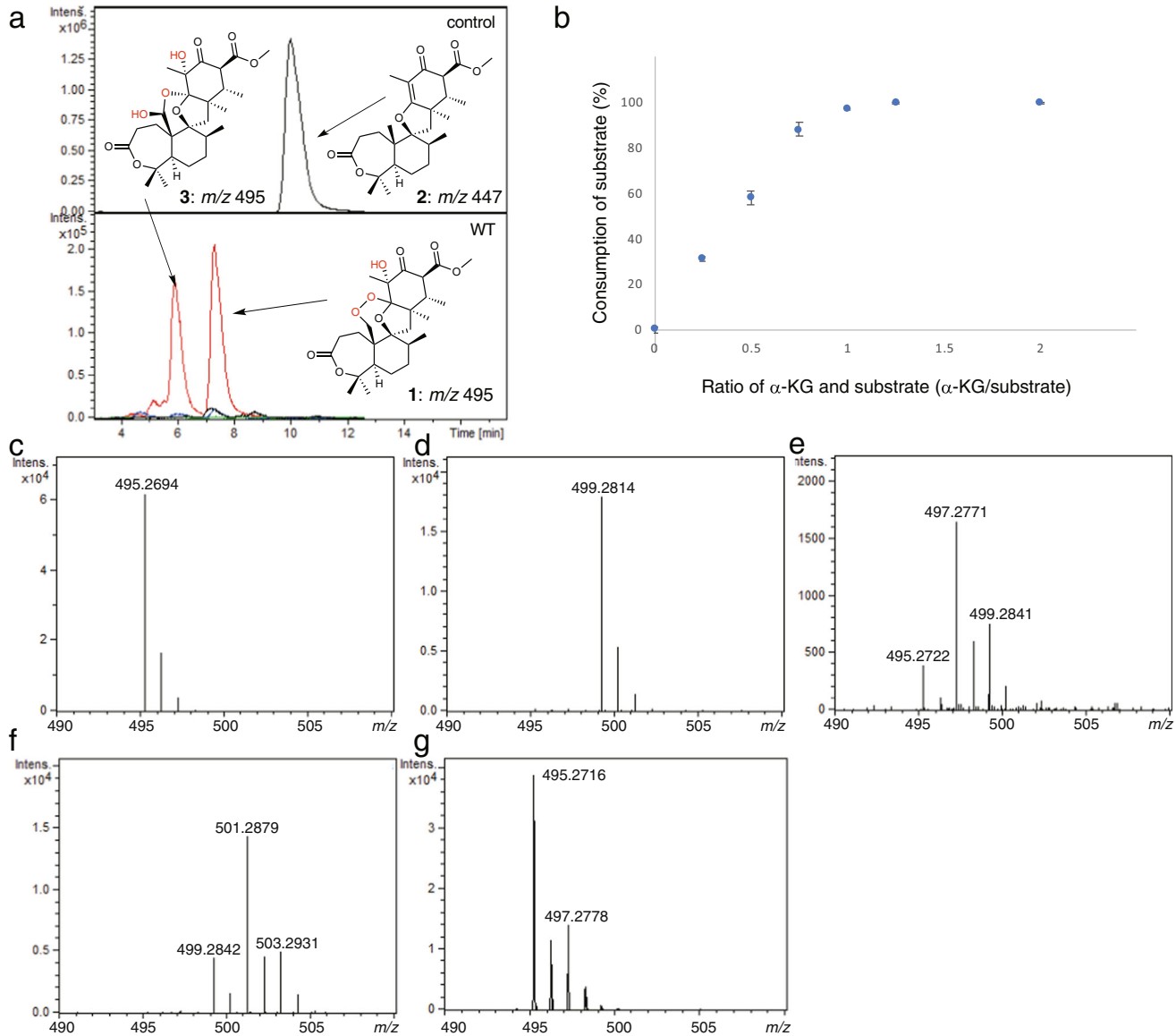

**Fig. 2 Enzyme reactions of NvfI. a** In vitro enzyme reaction of NvfI with **2**. **b** Consumption of **2** in the enzyme reactions of NvfI, using varied ratios of **2** and α-KG. **c–f** The $^{18}O_2$ and $H_2^{18}O$-labeling experiment. The enzyme reactions containing (**c**) $^{16}O_2$ and $H_2^{16}O$, (**d**) $^{18}O_2$ and $H_2^{16}O$, (**e**) $^{16}O_2$ and $H_2^{18}O$, and (**f**) $^{18}O_2$ and $H_2^{18}O$. **g** Overnight incubation of **1** in $H_2^{18}O$-containing buffer. All experiments were repeated independently more than three times with similar results. The blue dots are means of $n = 3$ independent experiments and error bars indicate standard deviations.

by non-enzymatic exchange via a hemiketal intermediate (Supplementary Fig. 4). Indeed, the +2 peak ($m/z$ 497) was observed when **1** was incubated in $H_2^{18}O$ without NvfI while the incorporation efficiency is relatively slow (Fig. 2g). The C-3' hydroxyl group in **1** was the other possible site of oxygen incorporation from water. Surprisingly, however, only 1.2% of +6 peak ($m/z$ 501) was observed in $^{18}O_2$-labeling experiment, which indicated that the oxygen atom of C3' hydroxyl group of **1** is not from $O_2$ molecule, but mostly from $H_2O$. While the hydroxyl group introduced by the action of Fe(II)/α-KG-dependent hydroxylases is usually derived from $O_2$, the oxygen ligands in both the ferryl and ferric states can be exchanged with the solvent water during the catalysis[40]. In the case of NvfI catalytic cycle, it is remarkable that almost all of the ferryl and/or Fe(III)-OH species exhibited oxo/water exchange prior to the formation of **1**. Alternatively, it would be also possible that water molecule is incorporated into the C-3' position of **2** through formation of a cationic intermediate.

**Overall structure of NvfI.** Interestingly, the gel-filtration analysis of NvfI revealed that the apparent molecular weight of NvfI is 34 kDa, indicating that NvfI exists as a monomer in solution (Supplementary Fig. 5). The formation of the monomeric state is notable, because other reported fungal αKG-dependent oxygenases, involved in fungal meroterpenoid biosynthesis, normally exist as homodimers[10,19,25,26]. Since functions and oligomeric forms of the other NvfI homologs (Supplementary Fig. 2) have not been studied, we do not know how widespread this feature is. Notably, isopenicillin N synthases and deacetoxycephalosporin C synthases are also monomeric in solution[34,41].

To understand the structural details of the NvfI-catalyzed endoperoxide-forming reaction, we solved the X-ray crystal structures of NvfI complexed with Fe, α-KG, and substrate **2** at 1.9 Å resolution. The overall structure of NvfI possesses a double-stranded β-helix core (DSBH) fold, which is highly conserved in the α-KG-dependent oxygenase family[9,10,19,26,41] (Fig. 3 and Supplementary Fig. 6a–c). The major structural differences

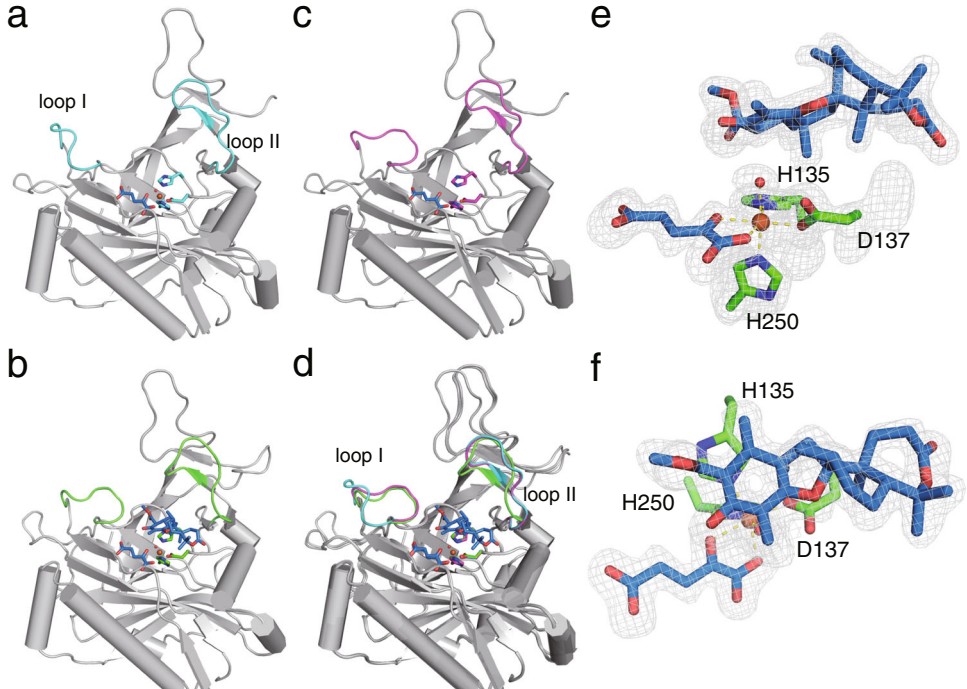

**Fig. 3 Crystal structure of NvfI. a–c** The overall structures of NvfI. **a** Structure of state I with α-KG, (**b**) structure of state II with NOG, and (**c**) structure of state III in complex with **2** and α-KG. **d** Superimposed view of (**a–c**). **e**, **f** Fo-Fc polder maps of the catalytic center of NvfI and substrates from different angles. The electron density map of the ligands is represented by a gray mesh, contoured at +3.0 sigma. The iron coordination is represented by dashed yellow lines. The loop I and loop II in state I–III were colored cyan, magenta, and green, respectively. **2**, α-KG, and NOG were depicted as blue sticks.

between NvfI and other α-KG-dependent oxygenases lie in the N-terminal and C-terminal regions. The N-terminus of NvfI consists of two anti-parallel β-strands (β1 and β3) connected by a long loop. The β1 and β3 strands are involved in the formation of a β-like structure together with three anti-parallel β-sheets (β4′, β5′, and β7′) in the DSBH fold, in remarkable contrast to other α-KG-dependent oxygenases[9,10,19,26,41] (Supplementary Fig. 6c–e).

**The active site architecture of NvfI.** The active site of NvfI is located in the DSBH core and an Fe is coordinated by the His135-His137-Asp250 triad, which is conserved in α-KG-dependent oxygenases (Fig. 3d, e). Interestingly, a comparison of the binary complex structure with α-KG (monomer B, state I) and the ternary complex structure with **2** and α-KG (monomer A, state III) revealed that the residues Ser122-Gly128 on loop I and Trp199-Pro209 on loop II undergo significant conformational changes upon substrate binding (Figs. 3, 4). The Ser122-Gly128 residues on loop I flip toward the active site, and Phe127 on the loop closes the substrate binding site along with the movement of Trp199 on loop II (Figs. 3, 4 and Supplementary Fig. 7). Furthermore, the Arg201 side chain moves toward Tyr116 and forms hydrogen bond networks among Tyr116, Arg201, Asp206, and water molecules. In contrast, Glu208 moves away from the active site to avoid steric hindrance with the substrate, and hydrogen bonds with Lys205 to support the conformation of loop II.

In addition, we also obtained a partially closed conformation of NvfI (monomer B, state II) from another data set of the wild type in complex with **2** and an α-KG analog N-oxalylglycine (NOG). The structural analysis revealed that the overall structure and the active site architecture of monomer A (complex structure with **2** and NOG) are almost identical to that of the ternary complex with **2** and α-KG (state III), while the loop I in monomer B (complex structure with NOG) takes a different conformation from that in the open state I and flips toward the active site

(Figs. 3, 4 and Supplementary Fig. 7). In this state, loop II takes the same open conformation as that in state I but the electron density of the side chains of Ile126 and Phe127 on the loop I was not clearly observed, indicating that the conformations of these regions are still not fixed. These observations suggested that conformational changes of the active site occur during the enzyme reaction to create a tunnel for the substrate and close the substrate-bound active site, although the density of **2** was not clearly observed in state II (Fig. 4c, d).

**The substrate binding mode.** In the complex structure of NvfI with Fe, α-KG, and **2**, the octahedral iron-center is coordinated by the bidentate α-KG and a water molecule, together with the catalytic triad (Fig. 3d, e). Substrate **2** directly interacts with Arg118 and His138 via hydrogen bonds at the C-4′ ketone and C-3 ester carbonyl groups of **2**, respectively. The C-9′ carbonyl group of the methyl ester also interacts with Thr133 and Arg132 via water molecules.

Notably, the distance between C-7′ of **2** and the iron-center (4.2 Å) is shorter than that between C-13 and the iron (6.5Å), which may explain the formation of the previously characterized side product **4** (Fig. 5)[12], even though the hydrogen atom at C-13 should be abstracted during the conversion to **1**. This conformation may represent a different stage of the catalysis or a crystallographic artifact. The docking model with **2**, based on the binary complex structure, suggested that the active site has enough space to bind the substrate in different conformations and that the C-13 position is located closer to the iron-center (Supplementary Fig. 8). Furthermore, a comparison of the substrate binding site in the binary and ternary complex structures revealed that the movement of the loop around Glu208 increases the active site volume around the A-ring moiety of **2**. These analyses suggested that the initial binding site of **2** is closer to loop I due to the steric hindrance between Glu208 and

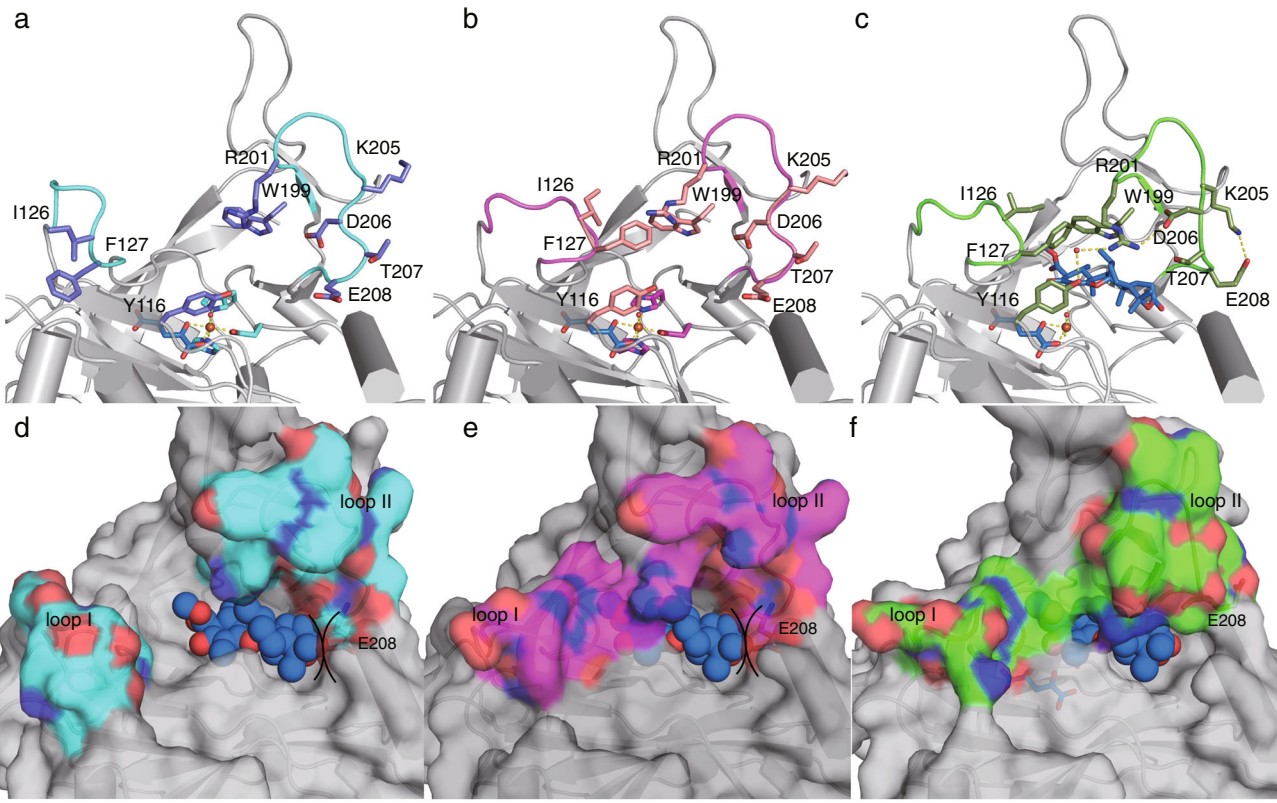

**Fig. 4 Active site structure of NvfI. a–c** The active site architectures of NvfI. **a** State I with α-KG, (**b**) state II with NOG and (**c**) state III in complex with **2** and α-KG. Dashed yellow lines represent hydrogen bonds. **d–f** Surface views of the active site of NvfI, (**d**) State I with α-KG, (**e**) state II with NOG and (**f**) state III in complex with **2** and α-KG. The ligand was superimposed in the α-KG binding structures (state I and II) of NvfI in panel (**d**) and (**e**) to show the steric hindrance between the substrate and Glu208.

the substrate (Fig. 4c–e), and the radical formation at C-13 and subsequent conformational changes of the active site move the substrate radical deeper inside the tunnel, as observed in the ternary complex structure.

**Mutagenesis experiments of NvfI.** Phe127 and Trp199 in loops I and II, which undergo dramatic conformational changes, may function as gatekeeping residues to load the substrate into the active site. Moreover, the flipping of Glu208 on loop II creates additional space for binding the A-ring moiety of **2** (Fig. 3). To investigate the importance of these conformational changes, variants of Phe127, Trp199, and Glu208 were constructed (Figs. 5c and 6). The substitution of Phe127 and Trp199 with Ala completely abolished the enzyme activity, while the E208A variant increased the activity to 144%. Furthermore, the substitution of Phe127 with Ile and Trp199 with Phe dramatically reduced the endoperoxide-forming activity, while also forming **4** and a product with m/z 461 (total activities were 6.7 and 25%, respectively). The product was determined to be the C-13-aldehyde **5**, which is generated via two rounds of two-electron oxidation of **2** (Supplementary Fig. 9a and Supplementary Table 3).

We solved the ternary complex structure of the NvfI W199F variant with **2** and NOG at 2.3 Å. Although the overall structure is almost identical to that of the wild type enzyme (rmsd value of 0.8 Å for Cα-atom), a comparison of the two structures revealed that the space around the carboxyl ester of D-ring and the active site entrance is significantly increased due to the substitution of the large indole ring to a smaller benzene ring (Supplementary Fig. 10). As a result, the D ring of substrate moves ~1 Å toward the 199 position, and the substrate is not suitably accommodated

within the deeper part of the tunnel, while the C-13 position is still close to the iron-center to generate compound **5**. In contrast, the increased activity of the E208A variant would be due to the decreased steric hindrance between the carboxylic acid of Glu and the C-14-C-15 dimethyl moiety and the clash during the conformational change of loop II. These observations suggested that after hydrogen abstraction from the C-13 methyl group, the bulky aromatic rings of Phe127 and Trp199 shift the substrate radical toward Glu208, thereby blocking the hydroxylation at the C-13 position.

A Tyr residue plays a crucial catalytic role in the reactions of COX and FtmOx1[6–10]. In NvfI, only one Tyr116 is located in the active site (Fig. 4b). This residue forms part of the active site around the B- and C-rings of the substrate. To test whether Tyr116 is involved in the NvfI catalysis, it was substituted with Ala and Phe residues (Figs. 5c and 6). Interestingly, the substitution of Tyr116 with Ala and Phe still maintained 44% and 101% activities, respectively. The kinetic analysis revealed that the catalytic specificity ($k_{cat}/K_M$) of the Y116A variant was not significantly altered (1.2-fold lower than the wild-type), while the $K_M$ and $k_{cat}$ values toward **2** were decreased by 1.3-fold and by 1.7-fold, respectively, as compared to the wild-type enzyme (Supplementary Fig. 11). The slight decrease in the $k_{cat}$ value would be caused by the decrease in the contacts between the enzyme and product. These results indicated that the Tyr116 residue does not participate in the catalytic mechanism of NvfI, which is clearly distinct from those of COX and FtmOx1.

To identify the catalytic residues, other candidates such as Ser114, His138, and Arg201, located close to **2**, were substituted with Ala. As a result, the substitution of Ser114 and Arg201 with

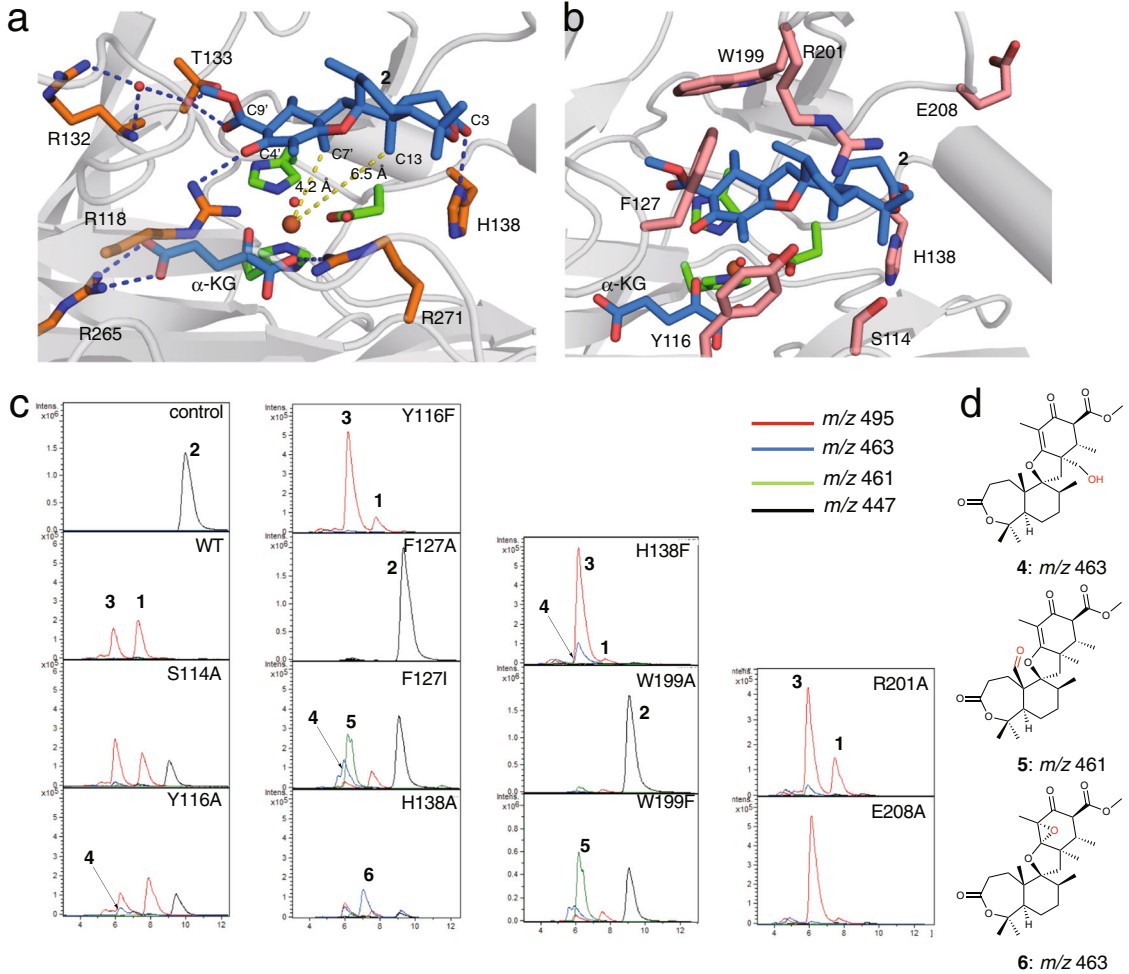

**Fig. 5 The binding modes of 2 and α-KG in the active site of NvfI and enzyme reactions of NvfI variants. a** Close-up view of the active site of NvfI in complex with **2** and α-KG. **b** The mutated residues in the active site of NvfI. The substrates **2** and α-KG are depicted by magenta stick models. The catalytic triad is represented by a green stick model. Dashed blue lines represent hydrogen bonds. The dashed yellow line shows the distance between C7′-Fe and C13-Fe. Water molecules are represented by red nb_spheres. **c** LC-MS charts of the enzyme reactions of NvfI variants. **d** Structures of shunt products.

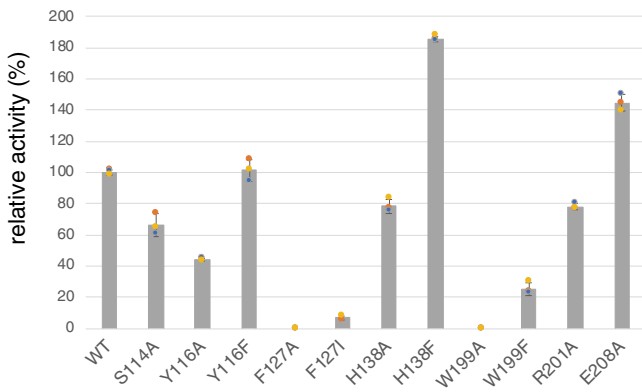

**Fig. 6 Graph representing the relative activities of the NvfI variants.** The bars are means of $n = 3$ independent experiments and error bars indicate standard deviations. Data are presented as mean values ± SD. All experiments were repeated independently three times with similar results. The relative activities were calculated from the total turnover number.

Ala modestly reduced the **1**-forming activity (66% and 78%, respectively) (Figs. 5c and 6). Interestingly, the substitution of His138 with Ala reduced endoperoxide-forming activity, and generated the previously identified **4**[12] and a product with $m/z = 463$ as major products. NMR analyses identified the product as the C-2′-C-3′ epoxide **6** with the (2′S,3′R) configuration (Supplementary Fig. 9b and Supplementary Table 2). In contrast, the product profile of the H138F variant reaction was similar to that of the wild-type, while the production of **4** was increased (total activities were 184%) (Figs. 5 and 6). The initial binding mode of the substrate would be altered, due to the enlargement of the active site by the large-to-small substitution of His138 with Ala (Supplementary Fig. 12). The substrate may bind in the deeper part of the active site toward α2, and the C-2′-C-3′ double bond is closer to the iron-center, resulting in the formation of **6**. Thus, the interaction between His138 and the C-3 ester group of **2** is critical for the hydrogen atom abstraction from C-13, to initiate the endoperoxide formation leading to **1**. Taken together, these results suggested that NvfI does not utilize any active site residues to mediate the abstraction or donation of a hydrogen atom in the catalytic mechanism, in sharp contrast to the well-studied endoperoxidases COX and FtmOx1[6–10].

## Discussion

Despite the many synthetic and biosynthetic approaches for the production of structurally unique and biologically active endoperoxide-containing compounds[2,3,6–10,42], the detailed bio-synthetic mechanisms for the endoperoxide-formation reactions still remain to be elucidated. In some cases, non-enzymatic reactions with singlet oxygen ($^1O_2$), generated through light absorption by photosensitizers, have been proposed (e.g., for the production of the antimalarial drug artemisinin in plants)[3,43–45]. On the other hand, in the previously proposed mechanisms for the enzymatic formation of endoperoxides by COX and FtmOx1[6–10], the active site Tyr residues play critical roles as a hydrogen atom transfer intermediary in COX and a hydrogen atom donor in FtmOx1, respectively. In contrast, our present mechanistic and structural characterizations of NvfI indicated that NvfI does not utilize any active site residues to generate amino acid-based radical intermediates in the endoperoxide-forming reaction. Thus, NvfI employs a different mechanism from those of the COX and FtmOx1 enzymes.

Several pathways can be proposed for the NvfI-catalyzed conversion of **2** to **1** (Supplementary Fig. 13). The ferryl species may abstract a hydrogen atom at C-13 of **2** to generate the pri-mary radical **A**, which then reacts with $O_2$ to form a peroxyl radical **B**. The O-centered radical may undergo radical addition at C-2' and the resulting radical **C** may receive the hydroxyl group from the iron(III)-hydroxyl species to yield **1** (Path 1). It is also possible that hydrogen atom abstraction or donation is mediated by an amino acid residue of NvfI as the COX-like or FtmOx1-like mechanisms. Alternatively, **C** may be reduced by a reductant (such as ascorbate under the in vitro condition) to generate **1** via **D** (Path 2). Other possible routes are generation via epoxide intermediates (Supplementary Fig. 13b). In path 2-4, NvfI thus requires two equivalents of α-KG. Our stoichiometric analysis of α-KG strongly suggested that only one α-KG is required for the production of **1**, which is consistent with the path 1. Conse-quently, any reaction pathways involving two equivalents of α-KG per **1** formation appear to be unlikely.

In the X-ray crystallographic analysis of NvfI, we obtained three different conformations of active site (state I–III). These conformational changes of the loops between Ser122-Gly128 and Trp199-Pro209 in the crystal structures suggested dynamic movement of active site during the enzyme reaction. Based on our structural and functional analyses, we propose a unique mechanism for the NvfI-catalyzed endoperoxide-forming reac-tion, as follows (Fig. 7). The enzyme reaction is initiated by loading substrate **2** into the active site of NvfI in the state I. Then,

**Fig. 7 Proposed mechanism of the NvfI enzymatic reaction.** Schematic representation of the proposed mechanism for the NvfI-catalyzed endoperoxide formation reaction.

the conformation of active site changes to state II via movement of loop I upon substrate binding. In this binding mode, C-13 of **2** is located close to the iron-center due to steric hindrance between the A-ring of substrate and Glu208, and then the direct hydrogen abstraction from the C-13 position by the ferryl species occurs. Before the hydroxyl rebound at C-13 with the ferric-hydroxy species, further conformational changes of the residues on loop II, especially flipping of Glu208, together with the movement of Phe127 and Trp199 could relocate the radical intermediate toward the inside of the tunnel to prevent the reaction between the C-13-radical and the ferric-hydroxy species. Instead, the C-13-radical reacts with $O_2$, which may enter the tunnel in the proximity of the A-ring moiety of the substrate. The resulting peroxyl radical attacks C-2' to form an endoperoxide-containing C-3'-radical intermediate. Since the ferric-hydroxy and C-3' become closer by the relocation of the intermediate, the radical is finally quenched via the unusual hydroxy-rebound from the ferric-hydroxy species to produce **1**. Alternatively, it would be also possible that one electron is directly transferred to the ferric iron to generate the C-3' carbocation intermediate, which is then hydrolyzed to yield **1**. The stereochemistry of the hydroxy-rebound or the addition of water molecule is also restricted to the *R*-configuration by the binding mode of the substrate in the active site. Further studies are required to completely conclude the dynamic conformational changes of active site and repositioning of the radical intermediates.

The Fe(II)/α-KG-dependent enzymes catalyze various types of reactions, including desaturation, epimerization, C-C bond formation, halogenation, ring expansion, and other skeletal rearrangements[29–32,46]. In these reactions, the enzymes control the outcome by suppressing the direct oxygen rebound between the substrate radical and the ferric-hydroxy species[40]. A similar mechanistic scenario with a long-lived ferric state may transpire in the NvfI-catalyzed endoperoxide formation, consistent with our $^{18}O$ labeling experiments in which the solvent-derived oxygen atom is also incorporated into the hydroxyl group at C-3'. In the case of NvfI, the elusion of the radical intermediate from the ferric-hydroxy intermediate, facilitated by the large conformational changes in the active site, would extend the lifetime of the intermediate to prevent direct oxygen rebound, thus facilitating the reaction with molecular oxygen to generate the endoperoxide.

In conclusion, our structure-function analyses of NvfI have suggested its unique mechanism for endoperoxide formation, which is distinct from the previously proposed mechanisms for the COX- and FtmOx1-catalyzed reactions. Future discoveries of endoperoxide-forming enzymes will provide further insights into the mechanistic details of rare endoperoxide-formation reactions in nature.

## Method

**General remarks**. Oligonucleotide primers were purchased from Eurofins Genomics. Other chemicals were purchased from Wako Chemical Ltd. (Tokyo, Japan). Analytical grade solvents were purchased from Kanto Chemical Co., Inc. (Tokyo, Japan). The HRMS data were obtained by using a compact microTOF-MS (Bruker) attached to an LC-20AD UHPLC system (Shimadzu) with a COSMOSIL 2.5C18-MS-II column (2 mm i.d. × 75 mm; Nacalai Tesque, Inc.). Analytical and semi-preparative HPLC runs were performed on a Shimadzu LC20-AD HPLC system, using a COSMOSIL C18-MS-II analytical column (4.6 × 250 mm, 5 μm) and a COSMOSIL C18-MS-II semi-preparative column (10.0 × 250 mm, 5 μm).

**Expression and purification of recombinant NvfI and variants**. The pET28a vectors containing the NvfI gene were prepared previously[12]. The plasmid was transformed into *Escherichia coli* Rosetta™ 2(DE3) pLysS and used for the expression of NvfI and its variants. The cells harboring the plasmids were cultured at 37 °C to an $OD_{600}$ of 0.6 in Luria-Bertani medium, containing 50 mg ml⁻¹ kanamycin and 34 mg ml⁻¹ chloramphenicol. Isopropyl β-D-thiogalactopyranoside (IPTG) was then added to 0.2 mM (final concentration) for target protein expression, and the cultures were continued for 20 h at 16 °C. All purification steps were performed at 4 °C. The cultured cells were resuspended in

50 mM Tris–HCl buffer, pH 7.5, containing 5% (v/v) glycerol, 100 mM NaCl, and 5 mM imidazole (buffer A). The cells were lysed by sonication and the insoluble debris was removed by centrifugation at 12,000 g for 40 min. The supernatant was loaded onto a Ni-NTA Resin (Thermo Fisher Scientific) column. The resin was washed with 50 column volumes of buffer A containing 20 mM imidazole, and then the NvfI was eluted with buffer A containing 300 mM imidazole. The protein solution was concentrated with Amicon Ultra-15 centrifugal filter devices (30 K MWCO, Millipore). The filtered protein solution was injected onto a MonoQ column (8 mL, GE Healthcare). The protein was eluted with a linear gradient of 50–1000 mM NaCl in 50 mM HEPES buffer (pH 7.6). The NvfI protein was further purified on a Superdex 200 pg column (GE Healthcare). The target protein was eluted with 20 mM Tris (pH 7.6) containing 50 mM NaCl, and concentrated to 21 mg ml⁻¹ with an Amicon Ultra-4 filter at 4 °C. The purity of the enzymes was monitored by SDS-PAGE. The protein concentrations were calculated by measuring the ultraviolet absorption at $A_{280}$[47].

**Crystallization and Structure Determination**. The NvfI crystals were obtained at 10 °C in 50 mM Tris-HCl (pH 7.0), containing 20% (w/v) PEG3350, 20 mM α-KG, 200 mM NaSCN, and 2 mM DTT, with 15 mg ml⁻¹ of purified NvfI, by the sitting-drop vapor-diffusion method. The initial model of NvfI was solved by single-wavelength anomalous diffraction method using anomalous signal of Zn. The NvfI crystals were incubated in the crystallization buffer containing 1 mM $ZnCl_2$ for 5 min at 20 °C to obtain the complex structure. To obtain ternary complex structure with **2** and α-KG, the crystals were moved in anaerobic Coy chamber and then the NvfI crystals were incubated in the crystallization buffer containing 1 mM **2** and 200 mM $FeSO_4$.

The NvfI W199F crystals and NvfI wild type in complex with **2** and N-oxalylglycine (NOG) were obtained at 10 °C in 50 mM HEPES (pH 7.5), containing 20% (w/v) PEG3350, 20 mM α-KG, 100 mM NaSCN with 20 mg ml⁻¹ of purified enzymes by the sitting-drop vapor-diffusion method. The cluster crystals were crushed and diluted 1000–10,000 times with crystallization buffer to use as seeds. Diffraction quality crystals were obtained in the buffer containing 50 mM HEPES (pH 7.5), 20% (w/v) PEG3350, 20 mM α-KG, 100 mM NaSCN with 5 mg ml⁻¹ of purified enzymes and seed solution (10% volume of drop). The crystal drops were 10 times diluted with crystallization buffer (without 20 mM α-KG) containing 20 mM NOG for overnight at 10 °C. Then, the crystals were transferred into the soaking buffer containing 50 mM HEPES (pH 7.5), 20% (w/v) PEG3350, 20 mM NOG, 100 mM NaSCN, 1 mM $FeSO_4$, and 20 mM **2**.

The crystals were transferred into the cryoprotectant solution (reservoir solution with 25% (v/v) glycerol), and then flash cooled at −173 °C in a nitrogen-gas stream. The X-ray diffraction data sets were collected at BL-1A (Photon Factory, Tsukuba, Japan), using a beam wavelength of 1.1 Å. The diffraction data sets for NvfI were processed and scaled using the XDS program package[48] and Aimless[49]. The determination of Zn sites and the generation of the initial model were performed with Crank2 in CCP4[50]. The initial phase of the NvfI complex structure was determined by molecular replacement, using NvfI-Zn as the search model. Molecular replacement was performed with Phaser in PHENIX[51]. The initial phase was further calculated with AutoBuild in PHENIX[52]. The NvfI complex structure was modified manually with Coot[53] and refined with PHENIX.refine[54]. The cif parameters of **2** for the energy minimization calculations were obtained by using the PRODRG server[55]. The final crystal data and intensity statistics are summarized in Supplementary Table 1. The Ramachandran statistics are as follows: 97.1% favored, 2.9% allowed for NvfI complexed with **2** and α-KG, 96.7% favored, 3.3% allowed for NvfI complexed with **2** and NOG, and 97.0% favored, 3.0% allowed for NvfI W199F complexed with **2** and NOG. A structural similarity search was performed, using the Dali program[56]. All crystallographic figures were prepared with PyMOL (DeLano Scientific, http://www.pymol.org). The structural analysis of NvfI complexed with **2** and NOG revealed that the overall structure and active site architecture of monomer A (complex structure with **2** and NOG) are almost identical to those of NvfI complexed with **2** and α-KG (state III) while the loop I in monomer B (complex structure with NOG) is different from that in state I. For the docking model, **2** was manually added in the active site, with Coot[53]. The structures were then calculated by PHENIX.refine[54] with simulated annealing.

**Preparation of NvfI substrates**. Compound **2** was obtained from the crude extract of the *Aspergillus novofumigatus* IBT 16806 *nvfI*Δ strain[12], grown on PDA plates for four weeks at room temperature. The metabolites were extracted with acetone, and the solvent was removed by evaporation. The crude extract was purified by silica gel chromatography (Wakogel C-200, 100% chloroform). The fraction containing **2** was collected and further purified with a Shimadzu HPLC system, using a TSK-gel ODS-80Tm column (Tosoh Co. Ltd., 7.8 mm i.d. × 300 mm, 43% acetonitrile, isocratic elution at 3.0 ml min⁻¹). The UV absorbance was monitored at 270 nm.

**Enzyme reaction of NvfI**. The standard enzyme reaction of NvfI was performed in 50 mM Tris-HCl buffer (pH 7.5), containing 100 μM **2**, 2 mM α-KG, 2 mM ascorbate, 200 μM $FeSO_4$, and 10 μM NvfI, for 8 h at 30 °C. The reaction was quenched by adding an equivalent volume of methanol. The samples were

centrifuged and clarified with a 0.22 μm filter, and the reaction products were analyzed by a compact microTOF-MS (Bruker) attached to an LC-20AD UHPLC system (Shimadzu). Isocratic elution was performed with 58% of $CH_3CN/H_2O$ solution both containing 0.1% formic acid. All measurements were conducted in triplicate.

**Stoichiometric analysis of NvfI**. To measure **2** consumption, the enzyme reaction of NvfI was performed in 50 mM HEPES buffer (pH 7.5), containing 100 mM **2**, 1 mM ascorbate, 20 μM $FeSO_4$, and 20 μM NvfI, and various concentration of **2** (0, 25, 50, 75, 100, 125, and 200 mM), for 30 min at 30 °C. The reaction was quenched by adding an equivalent volume of acetonitrile. The samples were centrifuged and clarified with a 0.22 μm filter, and the reaction products were analyzed by Prominence UHPLC system (Shimadzu). Isocratic elution was performed with 54% of $CH_3CN/H_2O$ solution both containing 0.1% formic acid.

To measure α-KG consumption, the enzyme reaction of NvfI was performed in 50 mM HEPES buffer (pH 7.5), containing 200 mM α-KG, 1 mM ascorbate, 20 μM $FeSO_4$, and 20 μM NvfI, and various concentration of **2** (0, 100, 150, 200, and 300 mM), for 30 min at 30 °C. The incubation mixture was filtered through a NANOSEP 10 K omega centrifugal device to remove protein. The samples were diluted 40 times with water and subjected to next α-KG detection reaction. The concentration of α-KG was quantified using α-KG detection kit (Dojindo) by following their protocol. Briefly, α-KG was converted into pyruvate using glutamic acid-pyruvic acid transaminase, and then the pyruvate was converted to $H_2O_2$ and acetyl phosphate with cascade enzyme reactions. The $H_2O_2$ was reacted with fluorescent dye by peroxidase and fluorescence of resorufin was measured using a microplate reader (Ex: 545 nm, Em: 590 nm).

**O₂ stoichiometry analysis**. The concentration of oxygen-saturated buffer was determined to be 1.3 mM according to published method[9].

Oxygen-saturated buffer (5 ml) was transferred into TERUMO syringe (10 ml) with a needle, and the syringe was then sealed by a rubber septum. An alkaline KI solution (2.1 M KI and saturated KOH dissolve in oxygen-free water) and 2.1 M $MnSO_4$ solution in oxygen-free water were prepared in anaerobic Coy chamber. 0.1 ml of alkaline KI solution and 0.1 ml of $MnSO_4$ solution were sucked into the syringe containing the 5 ml oxygen-saturated buffer. Then, the syringe was sealed again, quickly. The solution was mixed with gentle inversions of syringe. According to the following reaction, the $Mn(OH)_3$ precipitate formed.

$$4Mn^{2+} + O_2 + 8OH + 2H_2O \rightarrow 4Mn(OH)_3 (precipitation) \quad (1)$$

Wait 60 min to complete the reaction, and then $H_2SO_4$ solution (0.2 ml, 2.7 M) was sucked into the syringe, and $Mn^{3+}$ ions oxidized iodide to iodine under acidic conditions according to the following reactions:

$$2Mn(OH)_3(s) + 3H_2SO_4 \rightarrow 2Mn^{3+} + 3SO_4^{2-} + 3H_2O \quad (2)$$

$$2Mn_{3+} + 2I^- \rightarrow 2Mn^{2+} + I_2 \quad (3)$$

Iodine eventually formed $I_3$ ions with the excess KI:

$$I_2 + 2I^- \rightarrow I^{3-} (yellowish\ brown\ color) \quad (4)$$

The resulting iodine solution was transferred to a sample bottle and immediately titrated with standardized 2.5 mM $Na_2S_2O_3$ solution:

$$I^{3-} + 2S_2O_3^{2-} \rightarrow 3I^- + S_4O_6^{2-} \quad (5)$$

According to reactions (1)–(5), one equivalent of oxygen molecule corresponds to four equivalents of $Na_2S_2O_3$. Therefore, the oxygen concentration in the oxygen-saturated buffer was determined based on the amount of standardized 2.5 mM $Na_2S_2O_3$ solution used for titration. As a result, when I added 10.4 ml of 2.5 mM $Na_2S_2O_3$ solution, the solution became completely colorless. Therefore, the concentration of $O_2$ in air-saturated buffer is calculated to be 1.3 mM. The experiment was repeated twice.

To determine oxygen stoichiometry, oxygen-saturated buffer (final concentration 50, 100, 150, 200, 300 μM) was added to a total 150 μl anaerobic reaction mixture containing 50 mM HEPES (pH 7.5), 150 μM of asnovolin A, 100 μM of NvfI, 50 μM $FeSO_4$, 1 mM ascorbate, and 1 mM of α-KG in the Coy chamber. After starting the reaction, the reaction mixtures were sealed and incubated for 1 h at room temperature. The reactions were quenched by addition of equal amount of acetonitrile and the aggregated protein was removed by centrifugation at 20,400 g for 10 min. The reaction products were analyzed by Prominence UHPLC system (Shimadzu). Isocratic elution was performed with 48% of $CH_3CN/H_2O$ solution both containing 0.1% formic acid. All measurements were conducted in triplicate.

**Labeling Experiment**. The 50 mL reaction solutions, containing 50 μM NvfI enzyme, 100 μM **2**, 2 mM α-KG, 2 mM ascorbate, 200 μM $FeSO_4$, and 50 mM Tris–HCl buffer (pH 7.5), were prepared under anaerobic conditions in 500 μL tubes. The lids of the tubes were removed, and the tubes were placed in 13.5 mL glass vials. The vials were covered with a septum and the air inside was removed with a syringe. Afterward, $^{18}O_2$ gas (98 atom%) or air was injected into the vial. For the $H_2^{18}O$ labeling experiment, 40 mL of $H_2^{18}O$ (97 atom%) was added for the preparation of 50 mL reaction solutions (final concentration of $H_2^{18}O$ is 78%). The

enzyme reactions were performed at 20 °C for 2 h, and the products were analyzed with the same method as that for the standard reaction products.

**Site-directed mutagenesis, expression, and purification**. The plasmids expressing the variant of NvfI (S114A, Y116A, Y116F, F127A, F127I, H138A, H138F, W199A, W199F, R201A, and E208A) were constructed with a QuikChange Site-Directed Mutagenesis Kit (Stratagene), according to the manufacturer's protocol. The primers for the preparation of the variants are listed in Supplementary Table 5. The expression, purification, and enzyme reactions of all variants were performed in the same manners as for the wild-type enzymes.

**Large-scale isolation of the enzymatic products by Semi-preparative HPLC**. In-vitro reaction was performed with a 100 μL of buffer containing 50 mM Tris-HCl buffer (pH 7.5), 100 μM **2**, 2 mM α-KG, 2 mM ascorbate, 200 μM $FeSO_4$, and 50 μM of enzyme, for overnight at 30 °C in 200 of Eppendorf tubes. The reaction products were extracted by ethyl acetate and evaporated to get rid of the solvent. The crude extract was redissolved by methanol and isolated by semi-preparative HPLC using TSK-gel ODS-80TM column (Tosoh Co. Ltd., 7.8 i.d. × 300 mm, 43% acetonitrile, isocratic 3.0 ml min$^{-1}$). **5**: HR-ESI-MS $m/z$ $[M + H]^+$ 461.2532 (calc. 461.2534, $C_{26}H_{37}O_7^+$). **6**: HR-ESI-MS $m/z$ $[M + H]^+$ 463.2658 (calc. 463.2690, $C_{26}H_{39}O_7^+$).

**Steady-state enzyme kinetics**. To determine the time point for analyzing kinetic values, the consumption of substrate was measured at different time points. 0.1 μM of NvfI wild type or Y116A was incubated with 1.25 μM (for wild type) or 2.5 μM (for Y116A) of **2** in the buffer containing 50 mM HEPES (pH 7.5), 1 μM $FeSO_4$, 500 μM ascorbate, and 500 μM of α-KG at 30 °C for 1, 3, 5, and 10 min for wild type and 1, 3, 5, 10, and 20 min for Y116A. The reactions were quenched by addition of equal volume of acetonitrile and the aggregated protein was removed by centrifugation at 20,400 g for 10 min. The reaction products were analyzed by prominence UHPLC system (Shimadzu). All measurements were conducted in triplicate.

The kinetic values were determined with 2 min reaction, which is on pre-steady state. A concentration gradient of 1.2, 2.4, 4.8, 9.6, 19.2, 38.5, and 76.9 μM of substrate was used to determine the kinetic parameters of the wild type NvfI (0.1 μM). On the other hand, a concentration gradient of 1.25, 2.5, 5, 10, 20, 40, and 80 μM of substrate **2** was used to determine the kinetic parameters of NvfI-Y116A (0.1 μM). The reactions were performed in the buffer containing different concentration of **2**, 50 mM HEPES (pH 7.5), 0.1 μM of enzyme, 1 μM $FeSO_4$, 500 μM ascorbate, and 500 μM of α-KG at 30 °C for 2 min. Then, the reactions were quenched immediately by adding an equal volume of acetonitrile, using a syringe. The samples were centrifuged and clarified with a 0.22 μm filter, and the reaction products were analyzed by Prominence UHPLC system (Shimadzu). Isocratic elution was performed with 54% of $CH_3CN/H_2O$ solution both containing 0.1% formic acid. Curve fitting was performed with Prism 9. All measurements were conducted in triplicate.

**Reporting summary**. Further information on research design is available in the Nature Research Reporting Summary linked to this article.

## Data availability

The data generated in this study are provided in the Supplementary Information/Source Data file. The crystallographic data for the apo structures of NvfI wild type in complex with **2** and α-KG, NvfI wild type in complex with **2** and NOG, and W199F variant in complex with 2 and NOG have been deposited in the Protein Data Bank (PDB) under accession codes 7DE2 [57], 7ENB [58], and 7EMZ [59], respectively. All other relevant data are available from the corresponding author upon request. Source data are provided with this paper.

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

## Acknowledgements

The synchrotron radiation experiments were performed at the BL-1A of the Photon Factory. This work was supported in part by a Grant-in-Aid for Scientific Research from the Ministry of Education, Culture, Sports, Science and Technology, Japan (JSPS KAKENHI Grant Number JP16H06443, JP19K15703, JP20H00490, JP20KK0173, and JP20K22700), the New Energy and Industrial Technology Development Organization (NEDO, Grant Number JPNP20011), the PRESTO and ACT-X program from Japan Science and Technology Agency, The Uehara Memorial Foundation, Astellas Foundation for Research on Metabolic Disorders, Takeda Science Foundation, and Noda Institute for Scientific Research.

## Author contributions

T.M. and I.A. designed the experiments. T.M. and R.Z. performed in vitro analysis and crystallization experiments. R.Z. performed structure determination of enzyme reaction products. T.M., R.U., Y.M., and I.A. analyzed the data. T.M., R.U., and I.A. wrote the paper.

## Competing interests

The authors declare no competing interests.
