## [Peer Review File · Nature Communications]

Reviewers' Comments:

Reviewer #1:

Remarks to the Author:

This manuscript is a detailed study of endoperoxide formation by the enzyme NvfI, which is an Fe(II)- α KG dependent enzyme. Whereas other endoperoxide forming enzymes are thought to use a tyrosyl radical to enable radical formation on the substrate prior to reaction with oxygen and then endoperoxide formation, the authors here propose that no Tyr in the active site accumulates a radical. This work described here includes determination that one equivalent of α KG is needed for the reaction; that two oxygens from $^{18}\text{O}_2$ are incorporated and two oxygens from H_2^{18}O are incorporated into the product (with one of these coming from solvent exchange); the unusual monomeric structure of NvfI; the 1.8 Å resolution structure (with a conserved core structure); conformational changes upon binding of α KG and 2 that close the active site. Mutagenesis highlights key residues Phe and Trp, along with an interesting role for a Glu, but the only Tyr is not important in this reaction, ruling out a Tyr radical as key to the reaction. The authors also provide detailed structures on a number of side products that accumulate in the variant enzymes.

The authors have provided a large body of work here, but I felt the mechanistic conclusions were somewhat rushed based on the available data, or perhaps the material was not presented in a way that allowed me to appreciate how they had elucidated the full reaction.

Here are some suggestions:

- (1) While $^{18}\text{O}_2$ incorporation into the endoperoxide is most likely, they should demonstrate its incorporation at this moiety with NMR or other supporting data. Additionally, they could provide oxygen stoichiometry data to support that only a single oxygen is consumed in the reaction cycle.
- (2) The mechanistic proposal could be valid, but some direct evidence of the sequence of events through spectroscopic studies would be useful. Failing that, the authors could consider carrying out computational studies, for example, molecular dynamics, to interrogate the likely sequence of events in the reaction, which could also provide support for assignment of specific residues in conformational changes.
- (3) The monomeric state is very unusual. Can the authors provide some insight into how widespread this feature is likely to be in this 'clade' or beyond; and why this protein is monomeric
- (4) A lot of the paper is about the isolation of shunt / side products, however, none of these are shown in the paper, and it is hard to keep track of where every variant is in the active site. A combined Figure 4/ Figure 5 including the molecules could allow the interested reader to understand the distribution of the products and how it may relate to active site architecture.
- (5) The abstract states that no amino acid radical accumulates; but unless I misunderstood, no data is provided to rule out any amino acid radical. The data only show that the Tyr is not essential.
- (6) The abstract also states that "dynamic conformational changes of the active site, following the hydrogen abstraction from the substrate, relocate the position of the intermediate to prevent direct hydroxylation." I again believe this is a hypothesis based on the fact that they see two conformations in their crystal structures (with and without substrate), with differences from the docking, and nothing can be stated definitively yet about the relocation of the substrate
- (7) The discussion on the 4 different pathways is too long-winded, esp. when they are able to immediately rule out 3 of these possibilities, leaving only 1 pathway. I think this type of discussion should be moved to the Discussion, where they can in more detail discuss the parts of the mechanism they finally provide, providing comments on the knowns/unknowns.
- (8) How do the authors know the iron is in the Fe(II) state in the structure? How does the enzyme avoid turnover with Fe(II), α KG, and 2 present in the active site?
- (9) The context of the previous isolation of compound 4 needs to be provided (or at least a citation) for the broad readership of Nature Communications
- (10) The difference between the experimental crystal structure and a docking model is not evidence that movement of the substrate is as described on page 8. Additional evidence is required for these claims.
- (11) There are a very large number of statements in this paper -- for example "When the large phenyl or indole ring was substituted with a smaller isobutyl or phenyl side chain, the substrate was not suitably accommodated within the deeper part of the tunnel during the conformational

changes, while the C-13 position was still close to the iron-center to generate compound 5" – that are not supported by data, but are instead plausible explanations for why they see the results on the variants. The authors would need to provide some data / analysis to support these statements; for example, they could crystallize the variants with 2 and analyze the active site.

This manuscript by Mori, et al. reports structural and mechanistic analysis of NvII, a new iron(II)- and 2-oxoglutarate-dependent (Fe/2OG) endoperoxide synthase from *Aspergillus novofumigatus*, which generates fumigatonoid A by incorporation of 3 (!) oxygen atoms – two in the peroxide moiety and a third by formal hydroxylation, putatively in a single step – in the pathway to novofumigatonin. This reaction is different in character from the more well-understood endoperoxide installation reaction of veruculogen synthase (FtmOx1), and the authors rationalize the outcome in terms of (i) the absence of an H•-donating Tyr residue to complete a simple endoperoxidation (as in FtmOx1) and (ii) movement of the substrate mid-cycle to accommodate a post-endoperoxidation "rebound" step to a carbon remote from the site of the initial ferryl-mediated H• abstraction. The data and ideas put forth in this paper are very interesting, and I would see it as suitable for *Nature Communications*. The paper is also very well written. It is composed in the proper scholarly style of mechanistic analysis, wherein possibilities are mapped and dispositive experiments formulated, executed and interpreted. I like this study a lot. I do, however, have several questions and concerns, which I think the authors should be required to address. I will raise my substantive scientific concerns first and follow these with a few stylistic suggestions.

Substantive

1. The mechanistic schemes require some "cleaning up," specifically with respect to the state of the cofactor (whether there is a water ligand depicted and in what color) and the presence or absence of a hydroxyl group on C7', which I understand is installed in a side reaction but shows up in, for example, state B in Figure S3, **panel a**. An example of the water ligand issue is found in state 1 of the same figure and panel; if the unusual rebound step has occurred, there should not be a red (meaning derived from O₂) water ligand, because it has transferred to the product.
2. I am concerned about the result of the ¹⁸O₂ incorporation analysis. The text implies an ¹⁸O-atom composition of 98%, but the product has 23% of the ¹⁶O mass? I can't see any plausible mechanism by which the origin of the peroxide would not be O₂ gas, and so I would assume that its fraction of ¹⁸O would have to match the source. I suppose there could have been contamination by atmospheric O₂, but this reaction is sufficiently unusual that I am uneasy making this assumption. The authors could confirm that the 23% of the product lacking any ¹⁸O results from contamination by also measuring the isotopic distribution of the succinate co-product, which should have 23% ¹⁶O. I view this control as rather important.
3. In the same vein, the small quantity of the +6 species is absolutely *crucial* to their mechanistic argument, as it shows that the third installed oxygen atom – the reason that the reaction is so novel – derives in a small fraction of events from O₂. This is strong evidence for the unusual-rebound mechanism. Without it, there would be other possibilities. How many times was this experiment performed, and what was the variance?

4. Given the history of the FtmOx1 story, I view it as important that the authors have shown electron-density "cages" around the prime substrate **2**. I would ask that they provide in Figure 3 or a supplementary figure at least one more view (i.e., rotated) of the complex and "cage". In addition, I would like to see this figure magnified, either in the main text or in an additional supplementary figure.

5. The explanation of the quantitative rate measurements (enzyme assays) suggests that only one timepoint was taken. I don't understand how this approach allowed the authors to determine initial velocities with any confidence? What assumptions did they make about the time-dependence of turnover over the 10 min of the reactions? They should explain why they believe a single point is sufficient to define a rate. This problem gives me extra concern, given how very modest the reported k_{cat} values are. These enzymes typically have values of k_{cat} in the 10^{-1} - 10^1 s^{-1} range.

6. The proposed (1) initial open binding, (2) initiation of chemistry, and (3) mid-cycle motion of (a) radical state(s) is quite unusual. I do have concerns that there might be a less elaborate explanation. But it is an exciting hypothesis, and the authors have honestly stated it as such. Perhaps they might comment on how unusual it would be for the reaction to initiate in an open (initial) substrate-binding mode and then undergo a fairly significant mid-cycle conformational change?

Stylistic

1. "Ferryl-oxo" is sometimes used, but it is improper. Harry Gray named the ferryl form of iron and intended the oxo as part of that designation. Only when one is referring specifically to the *oxo ligand itself* within the ferryl complex is it proper to specify "oxo." In referring to the ferryl complex as a whole, use of oxo is redundant.

2. First page of result section, toward bottom: "quired" should be "required"

3. The unusual structure at the N-terminus does not seem to me to be consistent with the description " β -barrel". It clearly is β -like structure, but there don't seem to be enough strands to make up a barrel.

4. The authors use the word "mutant" in a non-standard way to refer to an amino acid substitution. Sometimes, authors will refer to the protein variant with a residue substituted as a "mutant", but even this use is less good than calling the protein a "variant". When one is writing "The x  y substitution diminished activity..." "substitution" is the best word.

Reviewer #3:

Remarks to the Author:

The manuscript submitted by Mori, et al. describes an x-ray crystallographic and biochemical study on a novel Fe(II)/2OG dependent endoperoxidase, NvfI, which converts asnovolin A to Fumigatonoid A by introducing three oxygen atoms during the catalysis. The authors presented the first crystal structure of NvfI. The structure revealed an enzyme quaternary complex (NvfI-Fe-2OG-substrate) presented in one of the monomers and an enzyme ternary complex (NvfI-Fe-2OG) in another monomer. Together with the results on steady-state enzyme assays, ¹⁸O-labeling experiments as well as protein mutagenesis studies, the authors proposed a reaction mechanism of NvfI. Instead of using redox-active amino acids to assist the endoperoxidation reaction in heme-dependent COX enzyme and another Fe/2OG dependent FtmOx1 enzyme, NvfI may utilize dynamics of the substrate binding to regulate enzyme reactivity to support endoperoxidation outcome and suppress hydroxylation outcome. Although substrate-bound crystal structure of NvfI is novel, the overall experimental evidence does not support some of the main conclusions of the paper. Some key experiments are missing or the current presented data need more substantial explanation. Therefore, this reviewer does not support the acceptance of this manuscript for publication on Nat. Comm. The detailed comments are listed below:

- 1) The ¹⁸O-labeling experimental results are not sufficiently explained. These results are important to reveal reaction mechanism. First of all, no error bars were given for all these results. The use of 98% ¹⁸O₂ resulted in a product isotope distribution of 4%:73%:23% for $\Delta m/z = +6$, $+4$, and 0 products. Based on the proposed mechanism provided by the authors, it is difficult to understand why there is substantial $\Delta m/z = 0$ product. The authors should comment on this point. Also, the authors stated that more than 90% of the ferryl-oxo and/or ferric-hydroxo species exhibit oxo/water exchange based on their ¹⁸O-labeling experiment. However, the authors did not mention how this percentage was estimated. This reviewer does not think this is a straight-forward conclusion. The authors should provide more explanation.
- 2) The result presented in Figure 2b is one of the key results in deriving the proposed mechanism. However, the authors did not provide sufficient experimental details to support this result. This reviewer suspects that the section named "Enzyme reaction of NvfI" in the SI may be the experimental conditions used to generate Figure 2b. Yet, there is no information about the amount of substrate used. Also, no description of LC running conditions is provided. It is less likely the authors could simultaneously detect substrate and 2OG in the same LC running conditions and quantify them. Also, it is well known that Fe/2OG enzymes exhibit uncoupling, meaning that even without substrate, 2OG could be consumed with the presence of ascorbate, iron and O₂. Even with the presence of substrate, normally more 2OG consumption is observed than the substrate consumption. Sometimes, this uncoupling is significant. Here in Figure 2b, the result suggests that NvfI is 100% coupled. Can the authors give some explanation for this? Overall, the description of the result in Figure 2b is lack of many important details, therefore it is difficult to judge its validity.
- 3) For the crystallographic results, the authors should provide more electron density maps and the corresponding modelling quality. Currently, only one figure with electron density of the active center with the substrate is provided. It is difficult to judge the overall quality of the crystallographic results.
- 4) The authors claimed that the metal center revealed in the crystallographic results is iron. However, based on the crystallization conditions described in the SI, only ZnCl₂ was introduced into the crystallization buffer. Therefore, it is very unclear how the authors assigned the metal center to be iron. Did anomalous x-ray scattering was collected?
- 5) The authors concluded that the substrate is most likely bound to an open conformation in the initial C-H activation step at C13 position, then the dynamics of loop I and loop II may bring the activated substrate deep into the protein, therefore preventing immediate hydroxyl rebound to C13 and promoting endoperoxide formation. Although, this is an interesting mechanistic hypothesis. The authors did not provide any evidence to support this. The open conformation in the crystal structure does not have substrate. Although the author docked the substrate into this open conformation but docking model cannot be used solely to support such a conclusion. More evidence is needed. Although the authors did see some changes with the mutagenesis studies, these changes cannot be easily correlated to the possible altering of the substrate positioning. Therefore, this conclusion can only be viewed as a speculation. Key experimental data, which is

missing in the current manuscript, are needed to support it.

Reviewer 1:

This manuscript is a detailed study of endoperoxide formation by the enzyme Nvfl, which is an Fe(II)- α KG dependent enzyme. Whereas other endoperoxide forming enzymes are thought to use a tyrosyl radical to enable radical formation on the substrate prior to reaction with oxygen and then endoperoxide formation, the authors here propose that no Tyr in the active site accumulates a radical. This work described here includes determination that one equivalent of α KG is needed for the reaction; that two oxygens from $^{18}\text{O}_2$ are incorporated and two oxygens from H_2^{18}O are incorporated into the product (with one of these coming from solvent exchange); the unusual monomeric structure of Nvfl; the 1.8 Å resolution structure (with a conserved core structure); conformational changes upon binding of α KG and 2 that close the active site. Mutagenesis highlights key residues Phe and Trp, along with an interesting role for a Glu, but the only Tyr is not important in this reaction, ruling out a Tyr radical as key to the reaction. The authors also provide detailed structures on a number of side products that accumulate in the variant enzymes.

The authors have provided a large body of work here, but I felt the mechanistic conclusions were somewhat rushed based on the available data, or perhaps the material was not presented in a way that allowed me to appreciate how they had elucidated the full reaction.

We really appreciate the reviewer's critical and thoughtful comments.

Here are some suggestions:

(1) While $^{18}\text{O}_2$ incorporation into the endoperoxide is most likely, they should demonstrate its incorporation at this moiety with NMR or other supporting data. Additionally, they could provide oxygen stoichiometry data to support that only a single oxygen is consumed in the reaction cycle.

Many thanks for the comment. Yes, it is a good idea to demonstrate the $^{18}\text{O}_2$ incorporation into the endoperoxide by analyzing NMR isotope shifts, however we could not obtain enough amount of the labeled compound. Instead, we newly repeated the $^{18}\text{O}_2$ incorporation experiments under strictly anaerobic conditions to avoid contamination by atmospheric $^{16}\text{O}_2$. As a result, we clearly reconfirmed that out of the three oxygen atoms incorporated, two oxygens are from $^{18}\text{O}_2$ and the other one mostly from H_2O . We believe this result further supports the $^{18}\text{O}_2$ incorporation into endoperoxide. Please also see our response to Reviewer 2's comments (2) and (3).

We have also provided the oxygen stoichiometry data and analysis as follows (page 5).

"On the other hand, for measurement of oxygen stoichiometry, enzyme reactions were performed using different amount of O_2 . As a result, the production of 1 increased with increasing O_2 concentrations, and

reached plateau when the ratio was ca. 2 (Supplementary Figure 3b-c), which indicated that two equivalents of O₂ are required for the installation of three oxygen atoms onto 2.”

(2) The mechanistic proposal could be valid, but some direct evidence of the sequence of events through spectroscopic studies would be useful. Failing that, the authors could consider carrying out computational studies, for example, molecular dynamics, to interrogate the likely sequence of events in the reaction, which could also provide support for assignment of specific residues in conformational changes.

Many thanks for the comment. Since we do not have access to stopped-flow system, rapid freeze quench apparatus, and EPR facility, we contacted several experts in the field for collaboration on the spectroscopic and computational studies to obtain further evidence to support our mechanistic proposal, however, unfortunately, we could not make it at this difficult time of the COVID-19 situation. We expect to fulfill these studies by future collaborations, but we feel that the present result can stand alone in terms of substance and novelty as a communication of an important discovery.

Instead, we newly performed α KG and O₂ stoichiometry analysis, revision of ¹⁸O₂-incorporation experiment, revision of enzyme kinetics, and solved additional X-ray crystallographic structures of wild type (second conformation) and W199F variant. Although further spectroscopic and computational studies are still required to fully conclude our proposal, we believe these data supported our proposal and that the mechanism of Nvfl is unique and distinct from those of the COX- and FtmOx1-catalyzed reactions.

(3) The monomeric state is very unusual. Can the authors provide some insight into how widespread this feature is likely to be in this ‘clade’ or beyond; and why this protein is monomeric.

The α -KG-dependent oxygenase superfamily enzymes are known to exist in various oligomeric forms. While PhyH-like oxygenases involved in the meroterpenoids biosynthesis, including AndA, PrhA, AusE, and FtmOx1, form dimers, isopenicillin N synthases (IPNS) and deacetoxycephalosporin C synthases (DAOCS) are monomeric in solution (Supplementary Figure 2). Since the functions and oligomeric forms of the other Nvfl homologues have not been studied yet, we do not know how widespread this feature is.

We have added a brief explanation as follows (page 6).

“Since functions and oligomeric forms of the other Nvfl homologues (Supplementary Figure 2) have not been studied, we do not know how widespread this feature is. Notably, isopenicillin N synthases (IPNS) and deacetoxycephalosporin C synthases (DAOCS) are also monomeric in solution^{34,41}.”

(4) A lot of the paper is about the isolation of shunt / side products, however, none of these are shown in the paper,

We isolated and determined the structures of shunt products 5 and 6 from the F127I, W199F, and H138A variants in this study. According to the reviewer's comment, we have newly added the detailed isolation method for 5 and 6 in the SI (page S8). The structure of another shunt product 4 from the wild type enzyme was determined in our previous study, for which we have cited the reference.

and it is hard to keep track of where every variant is in the active site. A combined Figure 4/ Figure 5 including the molecules could allow the interested reader to understand the distribution of the products and how it may relate to active site architecture.

We are sorry for the confusion. According to the suggestion, we have combined Figures 4 and 5 as a new Figure 5. On the other hand, the old Figure 3 has been separated into new Figures 3 and 4, according to Reviewer 2's comment (4).

(5) The abstract states that no amino acid radical accumulates; but unless I misunderstood, no data is provided to rule out any amino acid radical. The data only show that the Tyr is not essential.

We have revised the text from "amino acid radical" to "tyrosyl radical".

(6) The abstract also states that "dynamic conformational changes of the active site, following the hydrogen abstraction from the substrate, relocate the position of the intermediate to prevent direct hydroxylation." I again believe this is a hypothesis based on the fact that they see two conformations in their crystal structures (with and without substrate), with differences from the docking, and nothing can be stated definitively yet about the relocation of the substrate.

Yes, we agree that nothing can be stated definitively yet about the relocation of the substrate/intermediates. It is true that we just observed the conformations in the crystal structures (with and without substrate).

We have revised the abstract as follows (page 2).

"Further, the crystallographic analysis revealed significant conformational changes of two loops upon substrate binding, which suggests a dynamic movement of active site during the catalytic cycle. As a result, Nvfl installs "three" oxygen atoms onto a substrate in a single enzyme turnover. Based on these results, we propose a mechanism for the Nvfl-catalyzed, unique endoperoxide formation reaction to produce fumigatonoid A."

(7) The discussion on the 4 different pathways is too long-winded, esp. when they are able to immediately rule out 3 of these possibilities, leaving only 1 pathway. I think this type of discussion should be moved to the Discussion, where they can in more detail discuss the parts of the mechanism they finally provide, providing comments on the knowns/unknowns.

According to the suggestion, we have moved these sentences to the Discussion (page 12-13).

(8) How do the authors know the iron is in the Fe(II) state in the structure? How does the enzyme avoid turnover with Fe(II), α KG, and **2** present in the active site?

*Many thanks for the critical comment. It is true that we do not know the iron is in the Fe(II) state. We did not collect anomalous x-ray scattering, but just soaked crystals with 200 μ M FeSO₄ and 1 mM substrate **2**. Therefore, we have decided just to mention "Fe" instead of "Fe(II)" throughout the manuscript.*

(9) The context of the previous isolation of compound **4** needs to be provided (or at least a citation) for the broad readership of Nature Communications

We have cited a reference. Please see our response to Reviewer 1's comment (4).

(10) The difference between the experimental crystal structure and a docking model is not evidence that movement of the substrate is as described on page 8. Additional evidence is required for these claims.

*We performed additional X-ray crystallographic analyses several times, however we could not obtain the complex structure in which the substrate is bound to the open conformation (state I) in the initial C-H activation step at C13 position. Instead, we newly solved a partially closed structure (state II), in which the loop I takes a different conformation from that in the open state I and flips toward the active site, while loop II takes the same open conformation as that in state I. Although the electron density of **2** was not clearly observed, superimposition of **2** into the active site of state II predicted steric hindrance between the A-ring of substrate and Glu208, which suggested a different mode of the substrate binding to active site. However, we would admit this data may be still weak and more evidence is needed to fully conclude the proposal.*

We have added explanations as follows (page 8).

*"In addition, we also obtained a partially closed conformation of Nvfl (monomer B, state II) from another data set of the wild type in complex with **2** and an α -KG analogue N-oxalylglycine (NOG). The structural analysis revealed that the overall structure and the active site architecture of monomer A (complex structure with **2** and NOG) are almost identical to that of the ternary complex with **2** and α -KG (state III), while the loop I in*

*monomer B (complex structure with NOG) takes a different conformation from that in the open state I and flips toward the active site (Figures 3 and 4, Supplementary Figure 7). In this state, loop II takes the same open conformation as that in state I but the electron density of the side chains of Ile126 and Phe127 on the loop I are not clearly observed, indicating that the conformations of these regions are still not fixed. These observations suggest that conformational changes of the active site occur during the enzyme reaction to create a tunnel for the substrate and close the substrate-bound active site, although the density of **2** was not clearly observed in state II (Figure 4c and d)."*

We have also modified the text as follows (page 9).

*"The docking model with **2**, based on the binary complex structure, suggested that the active site has enough space to bind the substrate in different conformations and that the C-13 position is located closer to the iron-center (Supplementary Figure 8)."*

We have modified the Discussion as follows (page 13).

*"The enzyme reaction is initiated by loading substrate **2** into the active site of Nvfl in the state I. Then, the conformation of active site changes to state II via movement of loop I upon substrate binding. In this binding mode, C-13 of **2** is located close to the iron-center due to steric hindrance between the A-ring of substrate and Glu208, and then the direct hydrogen abstraction from the C-13 position by the ferryl species occurs. Before the hydroxyl rebound at C-13 with the ferric-hydroxy species, further conformational changes of the residues on loop II, especially flipping of Glu208, together with the movement of Phe127 and Trp199 could relocate the radical intermediate toward the inside of the tunnel to prevent the reaction between the C-13-radical and the ferric-hydroxy species."*

(11) There are a very large number of statements in this paper -- for example "When the large phenyl or indole ring was substituted with a smaller isobutyl or phenyl side chain, the substrate was not suitably accommodated within the deeper part of the tunnel during the conformational changes, while the C-13 position was still close to the iron-center to generate compound **5**" – that are not supported by data, but are instead plausible explanations for why they see the results on the variants. The authors would need to provide some data / analysis to support these statements; for example, they could crystallize the variants with **2** and analyze the active site.

*According to the reviewer's suggestion, we newly solved the X-ray crystal structure of the Nvfl W199F variant in complex with **2** and NOG. While the overall structure is almost identical to that of the wild type enzyme, comparison of the two structures revealed that the space around the carboxyl ester of D-ring is significantly increased due to the substitution of the indole ring to a smaller benzene ring.*

We have added the data and analysis as follows (page 10).

*“We solved the ternary complex structure of the Nvfl W199F variant with **2** and NOG at 2.3 Å. Although the overall structure is almost identical to that of the wild type enzyme (rmsd value of 0.8 Å for C α -atom), comparison of the two structures revealed that the space around the carboxyl ester of D-ring and the active site entrance are significantly increased due to the substitution of the large indole ring to a smaller benzene ring (Supplementary Figure 11). As a result, the D ring of substrate moves ~1 Å toward the 199 position, and the substrate is not suitably accommodated within the deeper part of the tunnel, while the C-13 position is still close to the iron-center to generate compound **5**.”*

Reviewer 2:

This manuscript by Mori, et al. reports structural and mechanistic analysis of Nvll, a new iron(II)- and 2-oxoglutarate-dependent (Fe/2OG) endoperoxide synthase from *Aspergillus novofumigatus*, which generates fumigatonoid A by incorporation of 3 (!!) oxygen atoms – two in the peroxide moiety and a third by formal hydroxylation, putatively in a single step – in the pathway to novofumigatonin. This reaction is different in character from the more well-understood endoperoxide installation reaction of veruculogen synthase (FtmOx1), and the authors rationalize the outcome in terms of (i) the absence of an H•-donating Tyr residue to complete a simple endoperoxidation (as in FtmOx1) and (ii) movement of the substrate mid-cycle to accommodate a postendoperoxidation "rebound" step to a carbon remote from the site of the initial ferrylmediated H• abstraction. The data and ideas put forth in this paper are very interesting, and I would see it as suitable for Nature Communications. The paper is also very well written. It is composed in the proper scholarly style of mechanistic analysis, wherein possibilities are mapped and dispositive experiments formulated, executed and interpreted. I like this study a lot. I do, however, have several questions and concerns, which I think the authors should be required to address. I will raise my substantive scientific concerns first and follow these with a few stylistic suggestions.

Thank you very much. We really appreciate the reviewer's support, and critical and instructive comments.

Substantive

1. The mechanistic schemes require some "cleaning up," specifically with respect to the state of the cofactor (whether there is a water ligand depicted and in what color) and the presence or absence of a hydroxyl group on C7', which I understand is installed in a side reaction but shows up in, for example, state B in Figure S3, panel a. An example of the water ligand issue is found in state 1 of the same figure and panel; if the unusual rebound step has occurred, there should not be a red (meaning derived from O2) water ligand, because it has transferred to the product.

According to the suggestion, we have modified Figure S14 (previous Figure S3) and Figure 6.

2. I am concerned about the result of the $^{18}\text{O}_2$ incorporation analysis. The text implies an ^{18}O -atom composition of 98%, but the product has 23% of the ^{16}O mass? I can't see any plausible mechanism by which the origin of the peroxide would not be O_2 gas, and so I would assume that its fraction of ^{18}O would have to match the source. I suppose there could have been contamination by atmospheric O_2 , but this reaction is sufficiently unusual that I am uneasy making this assumption. The authors could confirm that the 23% of the product lacking any ^{18}O results from contamination by also measuring the isotopic distribution of the succinate co-product, which should have 23% ^{16}O . I view this control as rather important.

We repeated experiments to re-investigate the $^{18}\text{O}_2$ -incorporation, this time under strictly anaerobic conditions to avoid contamination by atmospheric $^{16}\text{O}_2$. As a result, we confirmed that the product had only 2% of the ^{16}O mass, instead of 23% in the previous experiment (Figure 2 and Supplementary Table 1).

We have presented the data and revised the text as follows (Page 5).

“When **2** was incubated with Nvfl under $^{18}\text{O}_2$ (98 atom $^{18}\text{O}\%$), a major +4 peak of **1** was observed in the production of **1** (+6 : +4 : 0 = 1.2% : 95.7% : 2.0%) (Supplementary Table 1). Interestingly, in the experiment with H_2^{18}O (78 atom $^{18}\text{O}\%$), a major +2 peaks was observed, along with a minor peaks of the unlabeled **1** and +4 (+4 : +2 : 0 = 17.8% : 64.4% : 17.8%) (Figure 2 and Supplementary Table 1). Furthermore, the double labeling with 78% H_2^{18}O and 98% $^{18}\text{O}_2$ generated +4, +6 and +8 products of **1** (+8 : +6 : +4 : +2 : 0 = 25.6% : 48.9% : 23.2% : 1.8% : 0.5%).”

Further, we also measured the isotopic distribution of the succinate co-product, and confirmed that the +2 mass of the succinate co-product was observed when we incubated succinate in reaction buffer containing H_2^{18}O in the absence of enzyme. This indicated that the carboxylate is somehow nonenzymatically exchanged with the water molecules in the reaction buffer.

Fig. LC-MS analysis of incubation of succinate in the reaction solution. (A) Incubation of 100 μM succinate in reaction solution, containing 40 mM HEPES (pH7.5), 2 mM ascorbate, 2 mM α -KG, and 200 μM FeSO_4 , without enzyme for overnight. (B) Incubation of 100 μM succinate in reaction solution, containing 40 mM HEPES (pH7.5), 2 mM ascorbate, 2 mM α -KG, and 200 μM FeSO_4 , 80% H_2^{18}O without enzyme for overnight.

3. In the same vein, the small quantity of the +6 species is absolutely crucial to their mechanistic argument, as it shows that the third installed oxygen atom – the reason that the reaction is so novel – derives in a small fraction of events from O₂. This is strong evidence for the unusual-rebound mechanism. Without it, there would be other possibilities. How many times was this experiment performed, and what was the variance?

*Many thanks for the thoughtful comment. We tested the reactions in triplicate and summarized the data in Figure 2 and Supplementary Table 1 with standard error. The result indicated that only 1.2% of +6 peak was observed in ¹⁸O₂-labeling experiment, which indicated that the oxygen atom of C3' hydroxyl group of **1** is not from O₂ molecule, but mostly from H₂O. In the case of Nvfl catalytic cycle, it is remarkable that almost all of the ferryl and/or Fe(III)-OH species exhibit oxo/water exchange prior to the formation of **1**. Alternatively, it would be also possible that water molecule is incorporated into the C-3' position of **2** through formation of a cationic intermediate (Figure 6).*

We have revised the text as follows (page 6).

*“Surprisingly, however, only 1.2% of +6 peak was observed in ¹⁸O₂-labeling experiment, which indicates that the oxygen atom of C3' hydroxyl group of **1** is not from O₂ molecule, but mostly from H₂O. While the hydroxyl group introduced by the action of Fe(II)/ α -KG-dependent hydroxylases is usually derived from O₂, the oxygen ligands in both the ferryl and ferric states can be exchanged with the solvent water during the catalysis⁴⁰. In the case of Nvfl catalytic cycle, it is remarkable that almost all of the ferryl and/or Fe(III)-OH species exhibit oxo/water exchange prior to the formation of **1**. Alternatively, it would be also possible that water molecule is incorporated into the C-3' position of **2** through formation of a cationic intermediate.”*

We have also revised the text as follows (page 13).

*“the radical is finally quenched via the unusual hydroxy-rebound from the ferric-hydroxy species to produce **1**. Alternatively, it would be also possible that one electron is directly transferred to the ferric iron to generate the C-3' carbocation intermediate, which is then hydrolyzed to yield **1**.”*

4. Given the history of the FtmOx1 story, I view it as important that the authors have shown electron-density "cages" around the prime substrate **2**. I would ask that they provide in Figure 3 or a supplementary figure at least one more view (i.e., rotated) of the complex and "cage". In addition, I would like to see this figure magnified, either in the main text or in an additional supplementary figure.

According to the suggestion, we have added a new panel d in Figure 3 to show the electron density of substrate from different angle. Further, we have separated this figure into new Figure 3 and 4 to magnify the active site. In addition, we have also added the supplementary Figure 7 to show the electron density of amino acid residues according to Reviewer 3's comment.

5. The explanation of the quantitative rate measurements (enzyme assays) suggests that only one timepoint was taken. I don't understand how this approach allowed the authors to determine initial velocities with any confidence? What assumptions did they make about the time-dependence of turnover over the 10 min of the reactions? They should explain why they believe a single point is sufficient to define a rate. This problem gives me extra concern, given how very modest the reported k_{cat} values are. These enzymes typically have values of k_{cat} in the 10-1-101 s^{-1} range.

*Many thanks for the instructive comment. Unfortunately, the substrate availability is limited due to its low yield from the culture of *A. novofumigatus*. Thus, we could not carry out time-dependent experiments to determine the initial velocities for each substrate concentration. Nonetheless, we repeated the experiments. This time, we first measured consumption of substrate at different time points (1, 3, 5, and 10 min for wild type, and 1, 3, 5, 10, and 20 min for Y116A) (figure below). As a result, we selected 2 min, instead of 10 min, for the analyses, and obtained reasonable k_{cat} values ($0.28 s^{-1}$ for wild type and $0.17 s^{-1}$ for Y116A) as the reviewer suggested.*

We have described the detailed experimental procedures for the kinetic measurement in the SI, and revised the data and text accordingly.

6. The proposed (1) initial open binding, (2) initiation of chemistry, and (3) mid-cycle motion of (a) radical state(s) is quite unusual. I do have concerns that there might be a less elaborate explanation. But it is an exciting hypothesis, and the authors have honestly stated it as such. Perhaps they might comment on how unusual it would be for the reaction to initiate in an open (initial) substrate-binding mode and then undergo a fairly significant mid-cycle conformational change?

We appreciate the reviewer's thoughtful and instructive comments.

Please see our responses to Reviewer 1's comments (2), (6), and (10).

Stylistic

1. "Ferryl-oxo" is sometimes used, but it is improper. Harry Gray named the ferryl form of iron and intended the oxo as part of that designation. Only when one is referring specifically to the oxo ligand itself within the ferryl complex is it proper to specify "oxo." In referring to the ferryl complex as a whole, use of oxo is redundant.

We have changed "ferryl-oxo species" to "ferryl species".

2. First page of result section, toward bottom: "quired" should be "required"

We have revised the text.

3. The unusual structure at the N-terminus does not seem to me to be consistent with the description "b-barrel". It clearly is b-like structure, but there don't seem to be enough strands to make up a barrel.

We have revised the text to " β -like structure".

4. The authors use the word "mutant" in a non-standard way to refer to an amino acid substitution. Sometimes, authors will refer to the protein variant with a residue substituted as a "mutant", but even this use is less good than calling the protein a "variant". When one is writing "The x  y substitution diminished activity..." "substitution" is the best word.

We have revised the text to "variant" or "substitution".

Reviewer 3:

The manuscript submitted by Mori, et al. describes an x-ray crystallographic and biochemical study on a novel Fe(II)/2OG dependent endoperoxidase, Nvfl, which converts asnovolin A to Fumigatonoid A by introducing three oxygen atoms during the catalysis. The authors presented the first crystal structure of Nvfl. The structure revealed an enzyme quaternary complex (Nvfl-Fe-2OG-substrate) presented in one of the monomers and an enzyme ternary complex (Nvfl-Fe-2OG) in another monomer. Together with the results on steady-state enzyme assays, ¹⁸O-labeling experiments as well as protein mutagenesis studies, the authors proposed a reaction mechanism of Nvfl. Instead of using redox-active amino acids to assist the endoperoxidation reaction in heme-dependent COX enzyme and another Fe/2OG dependent FtmOx1 enzyme, Nvfl may utilize dynamics of the substrate binding to regulate enzyme reactivity to support endoperoxidation outcome and suppress hydroxylation outcome. Although substrate-bound crystal structure of Nvfl is novel, the overall experimental evidence does not support some of the main conclusions of the paper. Some key experiments are missing or the current presented data need more substantial explanation. Therefore, this reviewer does not support the acceptance of this manuscript for publication on Nat. Comm.

We appreciate the reviewer's critical and thoughtful comments. In the revised manuscript, we have added α KG and O_2 stoichiometry analyses, measurement of stoichiometry, revision of $^{18}O_2$ -incorporation experiment, revision of enzyme kinetics, and solved additional X-ray crystallographic structures of wild type (second conformation) and W199F variant. We have also provided detailed explanation of experimental methods in the SI. Based on these results, and combined with our previous data, we have slightly modified our proposed mechanism. We now propose that the abstraction of C-13 hydrogen atom should occur after the conformational change of loop I as observed in state I and the newly obtained state II. Then, the conformational changes of Glu208 could isolate the radical intermediate toward the inside of the tunnel to prevent the reaction between the C-13-radical and the ferric-hydroxy species. Although further spectroscopic and computational studies are still required to fully conclude our proposal, we believe these data supported our proposal and that the mechanism of Nvfl is unique and distinct from those of the COX- and FtmOx1-catalyzed reactions.

We have modified our proposed reaction mechanism as follows (page 13).

*"The enzyme reaction is initiated by loading substrate **2** into the active site of Nvfl in the state I. Then, the conformation of active site changes to state II via movement of loop I upon substrate binding. In this binding mode, C-13 of **2** is located close to the iron-center due to steric hindrance between the A-ring of substrate and Glu208, and then the direct hydrogen abstraction from the C-13 position by the ferryl species occurs. Before the hydroxyl rebound at C-13 with the ferric-hydroxy species, further conformational changes of the residues on loop II, especially flipping of Glu208, together with the movement of Phe127 and Trp199 could relocate the radical intermediate toward the inside of the tunnel to prevent the reaction between the C-13-radical and the ferric-hydroxy species. Instead, the C-13-radical reacts with O_2 , which may enter the tunnel in the proximity of the A-ring moiety of the substrate. The resulting peroxy radical attacks C-2' to form an endoperoxide-containing C-3'-radical intermediate. Since the ferric-hydroxy and C-3' become closer by the relocation of the intermediate, the radical is finally quenched via the unusual hydroxy-rebound from the ferric-hydroxy species to produce **1**. Alternatively, it would be also possible that one electron is directly transferred to the ferric iron to generate the C-3' carbocation intermediate, which is then hydrolyzed to yield **1**."*

The detailed comments are listed below:

1) The ^{18}O -labeling experimental results are not sufficiently explained. These results are important to reveal reaction mechanism. First of all, no error bars were given for all these results. The use of 98% $^{18}O_2$ resulted in a product isotope distribution of 4%:73%:23% for $\Delta m/z = +6, +4,$ and 0 products. Based on the proposed mechanism provided by the authors, it is difficult to understand why there is substantial $\Delta m/z = 0$ product. The authors should comment on this point. Also, the authors stated that more than 90% of the ferryl-oxo and/or ferric-hydroxo species exhibit oxo/water exchange based on their ^{18}O -labeling experiment. However,

the authors did not mention how this percentage was estimated. This reviewer does not think this is a straight-forward conclusion. The authors should provide more explanation.

Please see our response to Reviewer 2's comments (2) and (3).

*We repeated the experiments to re-investigate the $^{18}\text{O}_2$ -incorporation, this time under strictly anaerobic conditions to avoid contamination by atmospheric $^{16}\text{O}_2$. As a result, we confirmed that the product had only 2% of the ^{16}O mass, instead of 23% in the previous experiment (Figure 2 and Supplementary Table 1). The result also indicated that only 1.2% of +6 peak was observed in $^{18}\text{O}_2$ -labeling experiment, which indicates that the oxygen atom of C3' hydroxyl group of **1** is not from O_2 molecule, but mostly from H_2O .*

2) The result presented in Figure 2b is one of the key results in deriving the proposed mechanism. However, the authors did not provide sufficient experimental details to support this result. This reviewer suspects that the section named “Enzyme reaction of Nvfl” in the SI may be the experimental conditions used to generate Figure 2b. Yet, there is no information about the amount of substrate used. Also, no description of LC running conditions is provided. It is less likely the authors could simultaneously detect substrate and 2OG in the same LC running conditions and quantify them. Also, it is well known that Fe/2OG enzymes exhibit uncoupling, meaning that even without substrate, 2OG could be consumed with the presence of ascorbate, iron and O_2 . Even with the presence of substrate, normally more 2OG consumption is observed than the substrate consumption. Sometimes, this uncoupling is significant. Here in Figure 2b, the result suggests that Nvfl is 100% coupled. Can the authors give some explanation for this? Overall, the description of the result in Figure 2b is lack of many important details, therefore it is difficult to judge its validity.

Many thanks for the instructive comment. According to the reviewer's suggestion, we have added the detailed experimental details in the SI (page S5). Further, we carefully re-investigated the 2/ α KG stoichiometric experiment. First, we incubated α KG with enzyme in reaction buffer, containing HEPES, ascorbate, and Fe, under aerobic condition “without substrate”. As a result, no significant consumption of α KG was detected (figure below).

Fig. The α KG consumption “with” and “without” Nvfl” are shown in blue and orange dot, respectively. Different concentrations of α KG (0, 50, 100, or 200 μM) was incubated in 50 mM HEPES buffer (pH 7.5), containing 1 mM ascorbate, 20 μM FeSO_4 , (and 20 μM Nvfl) for 30 min at 30°C. The reaction mixture was filtered through a NANOSEP 10K omega centrifugal device to remove protein. The samples were diluted 40 times with water and the α KG concentration was quantified using α KG detection kit (Dojindo).

Next, we tested enzyme reactions “with substrate” in different ratio of **2**/ α KG. This time, consumption of both **2** and α KG were measured. The substrate was consumed completely at 1:1 ratio while consumption of α KG was completed at substrate/ α KG = 0.8. We think this slight difference would be due to the degradation of the substrate during analysis, or error of the substrate concentrations. Despite the difference, the data strongly suggested that Nvfl require one equivalent of α KG to generate endoperoxide **1**.

We have added the data in Figure 2 and Supplementary Figure 3 and revised the text as follows (page 5).
“To confirm the stoichiometry of the co-substrates, *in vitro* enzyme reactions were performed using varied ratios of **2** and α -KG under aerobic conditions. As a result, **2** was almost fully consumed when the α -KG : **2** ratio reached ca. 1 : 1 (Figure 2b and Supplementary Figure 3a), which suggested that one equivalent of α -KG is consumed during the formation of **1**.”

We have also provided oxygen stoichiometry data. Please see our response to Reviewer 1’s comment (1).

3) For the crystallographic results, the authors should provide more electron density maps and the corresponding modelling quality. Currently, only one figure with electron density of the active center with the substrate is provided. It is difficult to judge the overall quality of the crystallographic results.

According to the suggestion, we have added Figures 3e, f and Supplementary Figure 7 to show the electron density maps of active site residues and ligands.

Please also see our response to Reviewer 2’s comment (4).

4) The authors claimed that the metal center revealed in the crystallographic results is iron. However, based on the crystallization conditions described in the SI, only ZnCl₂ was introduced into the crystallization buffer. Therefore, it is very unclear how the authors assigned the metal center to be iron. Did anomalous x-ray scattering was collected?

Many thanks for the critical comment. We did not collect anomalous x-ray scattering, but soaked crystals with 200 μ M FeSO₄ and 1 mM substrate **2**. We have added detailed experimental method for the assignment of metal center by crystal soaking in the SI (Page S3).

Please also see our response to Reviewer 1’s comment (8).

5) The authors concluded that the substrate is most likely bound to an open conformation in the initial C-H activation step at C13 position, then the dynamics of loop I and loop II may bring the activated substrate

deep into the protein, therefore preventing immediate hydroxyl rebound to C13 and promoting endoperoxide formation. Although, this is an interesting mechanistic hypothesis. The authors did not provide any evidence to support this. The open conformation in the crystal structure does not have substrate. Although the author docked the substrate into this open conformation but docking model cannot be used solely to support such a conclusion. More evidence is needed. Although the authors did see some changes with the mutagenesis studies, these changes cannot be easily correlated to the possible altering of the substrate positioning. Therefore, this conclusion can only be viewed as a speculation. Key experimental data, which is missing in the current manuscript, are needed to support it.

We appreciate the reviewer's critical comments. Since we do not have access to stopped-flow system, rapid freeze quench apparatus, and EPR facility, we contacted several experts in the field for collaboration on the spectroscopic studies to obtain further evidence to support our mechanistic proposal, however, unfortunately, we could not make it at this difficult time of the COVID-19 situation. We expect to fulfill these studies by future collaborations, but we feel that the present result can stand alone in terms of substance and novelty as a communication of an important discovery.

Instead, as mentioned above, we newly performed α KG and O_2 stoichiometry analysis, revision of $^{18}O_2$ -incorporation experiment, revision of enzyme kinetics, and solved additional X-ray crystallographic structures of wild type (second conformation) and W199F variant. Although further spectroscopic and computational studies are still required to fully conclude our proposal, we believe these data supported our proposal and that the mechanism of Nvfl is unique and distinct from those of the COX- and FtmOx1-catalyzed reactions.

Please also see our responses to Reviewer 1's comments (2), (6), and (10).

Reviewers' Comments:

Reviewer #1:

Remarks to the Author:

The manuscript has improved and I do not have additional suggestions.

Reviewer #2:

Remarks to the Author:

In reviewing this manuscript again, I find myself conflicted. The reaction is interesting, and the O₂/substrate and 2OG/product stoichiometries do seem to establish that one turnover installs 3 oxygen atoms, which is novel. I don't think that the structures reveal much about the mechanism. Indeed, fundamental expectations, such as the proximity of the hydrogen atom to be abstracted to the iron cofactor, are not borne out. The proposed motion of the substrate/intermediate during the reaction to prevent HO• rebound and enable the more elaborate outcome is an interesting hypothesis, but there isn't really any evidence for it. The oxygen-isotope tracer experiments fail to provide any evidence that the third oxygen is installed by radical coupling to a hydroxo ligand, leaving the nature of this step in doubt.

The authors misunderstood my request to measure the isotopic composition of the succinate co-product. The idea was that, given the low enrichment of ¹⁸O from ¹⁸O₂ that they saw in the products, the most likely explanation was contamination of their reactions by atmospheric O₂. The point was to confirm that conclusion by confirming the same level of enrichment in the succinate, which incorporates one O-atom from O₂ as part of the reaction. Instead, they improved the important result to show higher enrichment in the prime products, making the succinate measurement that I suggested less important. Still, their misunderstanding of the point of the measurement is a bit disappointing, and the fact that they did not properly consider the statistics of incorporation of solvent oxygen at 1-4 of the four positions of succinate is also disappointing.

On balance, I still think the paper warrants publication in Nature Communications because of the novelty of the reaction.

The authors should check their use of past and present verb tense. In addition, the construction "...the substitution of His138 with Ala exhibited reduced endoperoxide-forming activity..." is not standard or correct. The substitution causes that diminished activity, and the variant exhibits the diminished activity. The substitution does not exhibit the diminished activity.

Reviewer #3:

Remarks to the Author:

In this revised manuscript, the authors addressed most of the concerns of the reviewers from the initial submission and improved the overall quality of the manuscript. I have two additional comments, one is major and the other is minor:

The major comment: The authors improved the section discussing the ¹⁸O labeling experiments. However, there is still some inconsistencies. In particular, the authors described that when ¹⁸O₂ was used, the major product (~ 95%) had +4 mass increase, suggesting an incorporation of two ¹⁸O atoms instead of three. The authors reasoned that the installed hydroxyl group at C3' position could have been exchanged with the solvent at the ferryl or the Fe(III)-OH state before the OH rebound step. In addition, the authors demonstrated that the carbonyl group at C4' position can exchange with the solvent. Thus, if these are correct, one should expect that when the reaction was run under ¹⁸OH₂ condition, the major product should still be the one with +4 mass increase (one on C3' and the other on C4'). However, the results presented in the current manuscript showed that the major product had +2 mass increase when ¹⁸OH₂ was used. This is inconsistent with the explanation put forward to explain the ¹⁸O₂ results. Also, when the reaction performed under both ¹⁸OH₂ and ¹⁸O₂ conditions, the +8 peak should be the major product. However, the actual major product was the +6 peak. Could the authors further comment on these observations.

This point is important for the authors to conclude that in the overall reaction, there are three O atoms installed into the product in the NvfI reaction. Also, the authors stated in the main text that the purity of 18OH₂ used is 78%. However, in the SI, it was stated that the purity of 18OH₂ is 98%. Is it a typo? Finally, please add all the expected m/z values for various substrates/products as well as isotope-labeled products when possible in the manuscript and in the figures so that the readers can easily look up this information.

The minor comment: In the mutagenesis study section, the authors mentioned a couple of times that the activities of some protein variants was 140% or 184%, etc. How were these numbers calculated? Is it based on the total turnover number relative to the native enzyme, or some steady-state kinetic rate constants? Please provide a brief description.

Reviewer 1:

The manuscript has improved and I do not have additional suggestions.

We appreciate the reviewer's comment.

Reviewer 2:

In reviewing this manuscript again, I find myself conflicted. The reaction is interesting, and the O₂/substrate and 2OG/product stoichiometries do seem to establish that one turnover installs 3 oxygen atoms, which is novel. I don't think that the structures reveal much about the mechanism. Indeed, fundamental expectations, such as the proximity of the hydrogen atom to be abstracted to the iron cofactor, are not borne out. The proposed motion of the substrate/intermediate during the reaction to prevent HO• rebound and enable the more elaborate outcome is an interesting hypothesis, but there isn't really any evidence for it. The oxygen-isotope tracer experiments fail to provide any evidence that the third oxygen is installed by radical coupling to a hydroxo ligand, leaving the nature of this step in doubt.

We really appreciate the reviewer's critical and thoughtful comments. We agree that more evidence are still needed to answer to these questions. In order to confirm the position of H abstraction by the iron cofactor, we should prepare the site-specifically deuterated substrates by in vitro enzyme reactions. However, we could not obtain such labeled compounds easily because the biosynthetic pathway for the substrate is known to involve some integral membrane proteins which are difficult to purify as soluble enzymes. Furthermore, additional spectroscopic and computational studies are still required to obtain solid evidence to support our mechanistic proposal. We expect to fulfill these studies by future collaborations, but we feel that the present result can stand alone in terms of substance and novelty as a communication of an important discovery.

The authors misunderstood my request to measure the isotopic composition of the succinate co-product. The idea was that, given the low enrichment of ¹⁸O from ¹⁸O₂ that they saw in the products, the most likely explanation was contamination of their reactions by atmospheric O₂. The point was to confirm that conclusion by confirming the same level of enrichment in the succinate, which incorporates one O-atom from O₂ as part of the reaction. Instead, they improved the important result to show higher enrichment in the prime products, making the succinate measurement that I suggested less important. Still, their misunderstanding of the point of the measurement is a bit disappointing, and the fact that they did not properly consider the statistics of incorporation of solvent oxygen at 1-4 of the four positions of succinate is also disappointing.

We are sorry for the confusion. What we would like to response to your first question about succinate is as follows.

According to the reviewer's previous suggestion, we optimized the LC-MS conditions to detect succinate and analyzed the incorporation of ¹⁸O-atom into succinate. We confirmed the incorporation of ¹⁸O-atom into

succinate in $^{18}\text{O}_2$ atmosphere. However, we also noticed the incorporation of ^{18}O -atom into succinate in the enzyme reaction in the presence of H_2^{18}O but absence of $^{18}\text{O}_2$. Considering the generally accepted mechanism of Fe(II)/ α -KG-dependent oxygenases, water-derived ^{18}O -atom should not be incorporated into succinate co-product during the formation the Fe(IV)-oxo species. Thus, we assumed that the oxygen atoms in succinate could exchange with solvent water or water oxygen could be incorporated into succinate in the process of the non-enzymatic degradation of α -KG. Indeed, incubation of succinate in a solution containing 40 mM HEPES (pH7.5), 2 mM ascorbate, 2 mM α -KG, and 200 μM FeSO_4 , 80% H_2^{18}O without enzyme for overnight generated +2 mass of succinate co-product (but not +4, +6, and +8). Therefore, we thought that it is difficult to accurately calculate the incorporation rate of ^{18}O -atom into succinate in the enzyme reaction. Thus, we could not use succinate production for confirmation of the enrichment of ^{18}O -atom in the product and succinate. Instead, we optimized the reaction condition to decrease the contamination of atmospheric $^{16}\text{O}_2$. We are very sorry again for the insufficient explanation in the last response letter.

On balance, I still think the paper warrants publication in Nature Communications because of the novelty of the reaction.

Thank you very much. We really appreciate the reviewer's comment.

The authors should check their use of past and present verb tense. In addition, the construction "...the substitution of His138 with Ala exhibited reduced endoperoxide-forming activity..." is not standard or correct. The substitution causes that diminished activity, and the variant exhibits the diminished activity. The substitution does not exhibit the diminished activity.

We hired an English-speaking person to go through the text and corrected the grammatical errors and misspellings.

Reviewer 3:

In this revised manuscript, the authors addressed most of the concerns of the reviewers from the initial submission and improved the overall quality of the manuscript. I have two additional comments, one is major and the other is minor:

The major comment: The authors improved the section discussing the ^{18}O labeling experiments. However, there is still some inconsistencies. In particular, the authors described that when $^{18}\text{O}_2$ was used, the major product (~95%) had +4 mass increase, suggesting an incorporation of two ^{18}O atoms instead of three. The authors reasoned that the installed hydroxyl group at C3' position could have been exchanged with the solvent at the ferryl or the Fe(III)-OH state before the OH rebound step. In addition, the authors demonstrated that the carbonyl group at C4' position can exchange with the solvent. Thus, if these are correct, one should expect that when the reaction was run under $^{18}\text{O}_2$ condition, the major product should still be the one with +4 mass increase (one on C3' and the other on C4'). However, the results presented in the current manuscript showed that the major product had +2 mass increase when $^{18}\text{O}_2$ was used. This is inconsistent

with the explanation put forward to explain the $^{18}\text{O}_2$ results. Also, when the reaction performed under both $^{18}\text{OH}_2$ and $^{18}\text{O}_2$ conditions, the +8 peak should be the major product. However, the actual major product was the +6 peak. Could the authors further comment on these observations. This point is important for the authors to conclude that in the overall reaction, there are three O atoms installed into the product in the Nvfl reaction.

We really appreciate reviewer's critical and thoughtful comments. Actually, to emphasize the non-enzymatic incorporation of ^{18}O into the product with H_2^{18}O , we incubated it longer time in the H_2^{18}O solution than that of enzyme reaction. We are sorry for forgetting to add details in method section. As a proper control experiment, we set the incorporation reaction of H_2^{18}O into the product under the same conditions as the other labeling reactions. As a result, the incorporation rate of ^{18}O was about 26%, which is consistent with the enzyme reactions using H_2^{18}O and/or $^{18}\text{O}_2$. Therefore, we still think that the carbonyl oxygen at C4' position can exchange with the solvent but not so efficiently. Notably, we also carried out the enzyme reaction with H_2^{18}O for the same incubation time and confirmed that the major product was still +2 of product. Accordingly, we have revised the panel g of Figure 2.

Also, the authors stated in the main text that the purity of $^{18}\text{OH}_2$ used is 78%. However, in the SI, it was stated that the purity of $^{18}\text{OH}_2$ is 98%. Is it a typo?

We are sorry for the confusion. We added 40 μL of H_2^{18}O (97 atom%) in total 50 μL of reaction solution. Other components were prepared with H_2^{16}O . Therefore, the final concentration of H_2^{18}O was 78% in the enzyme reaction. We have added "final concentration" before 78 atom $^{18}\text{O}\%$ in the main text (page 5) and "final concentration of H_2^{18}O is 78%" in the method section (page S8).

Finally, please add all the expected m/z values for various substrates/products as well as isotope-labeled products when possible in the manuscript and in the figures so that the readers can easily look up this information.

According to the reviewer's suggestion, we have added expected m/z values in the manuscript (page 5) and Figures 2 and 5.

The minor comment: In the mutagenesis study section, the authors mentioned a couple of times that the activities of some protein variants was 140% or 184%, etc. How were these numbers calculated? Is it based on the total turnover number relative to the native enzyme, or some steady-state kinetic rate constants? Please provide a brief description.

We used the total turnover number to calculate relative activity. We have added the explanation in the legend of Supplementary Figure 9 (page S22) as follows.

"The relative activities were calculated from the total turnover number."

Reviewers' Comments:

Reviewer #3:

Remarks to the Author:

Thanks for the authors to further improve their manuscript. I suggest the authors to add a sentence on the slow solve exchange on the C4' position in the 18O labeling experiment section so that the readers could understand the conclusion of this section a bit easier. Other than this, I do not have any further comments. The manuscript is now suitable to be published on Nat. Commun.

Reviewer 3:

Thanks for the authors to further improve their manuscript. I suggest the authors to add a sentence on the slow solve exchange on the C4' position in the 18O labeling experiment section so that the readers could understand the conclusion of this section a bit easier. Other than this, I do not have any further comments.

The manuscript is now suitable to be published on Nat. Commun.

We appreciate the reviewer's comment.

We have added the explanation of solvent exchange on the C4' in main text as follows (page 6):

"Indeed, the +2 peak (m/z 497) was observed when 1 was incubated in H218O without Nvfl while the incorporation efficiency is relatively slow."